# SonicSim: A customizable simulation platform for speech processing in moving sound source scenarios

**Kai Li**[1,2]**, Wendi Sang**[1]**, Chang Zeng**[3]**, Runxuan Yang**[1]**, Guo Chen**[1] **& Xiaolin Hu**[1,2,4*]

1. Department of Computer Science and Technology, Institute for AI,
BNRist, Tsinghua University, Beijing 100084, China
2. Tsinghua Laboratory of Brain and Intelligence (THBI),
IDG/McGovern Institute for Brain Research, Tsinghua University, Beijing 100084, China
3. National Institute of Informatics, Japan
4. Chinese Institute for Brain Research (CIBR), Beijing 100010, China
`tsinghua.kaili@gmail.com`
`xlhu@tsinghua.edu.cn`

## Abstract

Systematic evaluation of speech separation and enhancement models under moving sound source conditions requires extensive and diverse data. However, real-world datasets often lack sufficient data for training and evaluation, and synthetic datasets, while larger, lack acoustic realism. Consequently, neither effectively meets practical needs. To address this issue, we introduce *SonicSim*, a synthetic toolkit based on the embodied AI simulation platform Habitat-sim, designed to generate highly customizable data for moving sound sources. SonicSim supports multi-level adjustments—including scene-level, microphone-level, and source-level—enabling the creation of more diverse synthetic data. Leveraging SonicSim, we constructed a benchmark dataset called *SonicSet*, utilizing LibriSpeech, Freesound Dataset 50k (FSD50K), Free Music Archive (FMA), and 90 scenes from Matterport3D to evaluate speech separation and enhancement models. Additionally, to investigate the differences between synthetic and real-world data, we selected 5 hours of raw, non-reverberant data from the SonicSet validation set and recorded a real-world speech separation dataset, providing a reference for comparing SonicSet with other synthetic datasets. For speech enhancement, we utilized the real-world dataset RealMAN to validate the acoustic gap between SonicSet and existing synthetic datasets. The results indicate that models trained on SonicSet generalize better to real-world scenarios compared to other synthetic datasets. Code is publicly available at `https://cslikai.cn/SonicSim/`.

## 1 Introduction

Speech separation (Cherry, 1953) and enhancement (Loizou, 2007) are classic tasks focused on extracting target speech from noisy audio. The release of large-scale synthetic datasets like LibriMix (Cosentino et al., 2020) and the DNS Challenge (Reddy et al., 2020) has propelled model advancements, with many methods (Chen et al., 2020a; Luo & Yu, 2023b) demonstrating cross-environment transferability in zero-shot settings. These SOTA methods have significantly benefited applications such as online meetings (Rao et al., 2021) and human-computer interaction (Yu & Chen, 2017). However, a significant acoustic mismatch between synthetic and real-world data persists, leading to suboptimal model performance in real-world scenarios (Sivaraman & Kim, 2022).

Previous efforts to bridge the gap between synthetic and real-world data have mainly focused on more accurate modeling of room reverberations. Datasets like WHAMR! (Maciejewski et al., 2020) and the DNS Challenge (Reddy et al., 2020) simulate room impulse responses (RIRs) using methods such as finite element analysis (Jarrett et al., 2012), ray tracing (Lopez-Hernandez et al., 2000), or the

---

[*]Corresponding author.

image source method (Luo & Yu, 2023a). However, from the perspective of RIR simulation, existing approaches face three major challenges: 1) **Occlusion of obstacles**: RIRs based on the image source method struggle with varying quantities and shapes of obstacles, particularly when objects obstruct the sound path between the source and microphone (Aralikatti et al., 2022). 2) **Complex room geometry**: Simulations often assume cuboid-shaped rooms (Southern et al., 2011), deviating from complex real-world environments, such as lecture halls with tiered seating. 3) **Complicated room surfaces and object materials**: Different materials have distinct acoustic properties, and accurately replicating scenes using multiple materials is challenging. Therefore, accurate simulation of RIRs remains a difficult problem.

Existing datasets are primarily collected from static sound sources (Cosentino et al., 2020; Maciejewski et al., 2020; Reddy et al., 2020), making them suitable for non-moving scenarios like online meetings. In contrast, moving sound source scenarios—where speakers move around while static microphones capture their speech—are more relevant to applications such as robotic navigation (Wang et al., 2004). However, capturing large-scale datasets for moving sound sources is more time-consuming, leading to a scarcity of such data, which limits the development and evaluation of speech separation and enhancement models in these tasks. Additionally, real-world datasets are constrained by physical environments and, once collected, their distributions are fixed, making them difficult to adjust flexibly and increasing the risk of models overfitting to specific datasets. Thus, despite their high research value, the lack of relevant datasets for moving sound source scenarios significantly hinders deeper exploration in this area.

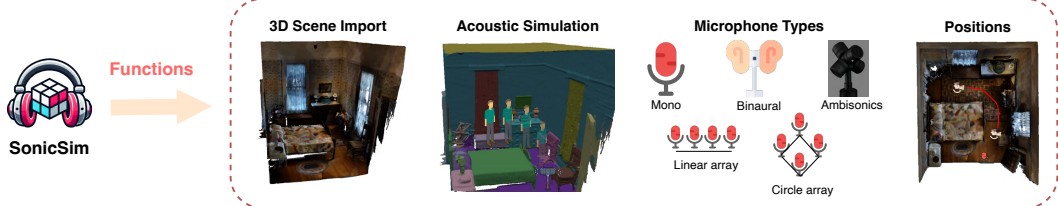

Figure 1: Overview of SonicSim, our toolkit for speech research. SonicSim provides a customizable data generator based on Habitat-sim, allowing users to generate realistic and physically plausible audio data in a controlled manner.

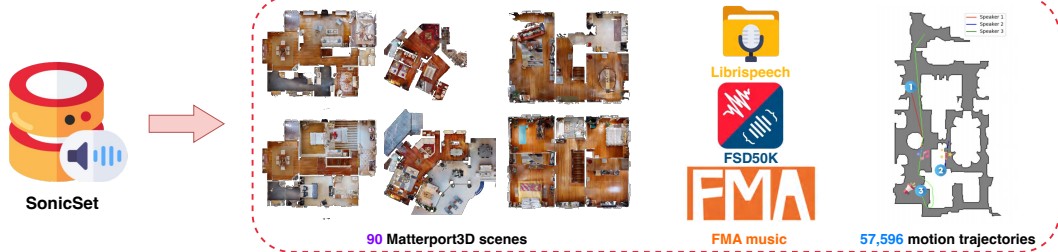

Figure 2: Overview of the SonicSet dataset. It covers a wide range of scenarios, speakers, and noise.

To advance research in moving sound sources, particularly in speech separation and enhancement tasks, there is a pressing need for a high-fidelity, low-cost, and comprehensive synthetic toolkit and data asset for moving sound sources. To address these issues, we developed *SonicSim*, a synthetic toolkit capable of accurately simulating RIRs for moving sound sources, as shown in Figure 1. This toolkit is built upon the embodied AI simulation platform, Habitat-sim (Savva et al., 2019), which can import a variety of 3D environments and accurately simulate the acoustic characteristics of rooms. The core strength of SonicSim lies in its precise simulation of RIRs (Chen et al., 2022a) and highly customizable simulation of moving sound sources, which significantly expands the scale of data collection for moving sound source research.

Based on SonicSim, we constructed a multi-scene, large-scale, and high-quality moving sound source dataset called *SonicSet* (see Figure 2): 1) **Multi-scene**: SonicSet utilizes 90 scenes from the Matterport3D dataset (Chang et al., 2017), covering a wide range of real-world environments, such as homes, offices, and churches; 2) **Large-scale**: SonicSet integrates 360 hours of speech audio from the LibriSpeech (Panayotov et al., 2015), combined with environmental noise from FSD50K

(Fonseca et al., 2021) and musical noise from the FMA dataset (Defferrard et al., 2017); 3) **High-quality**: The RIRs of the synthetic audio closely resemble real-world environments by simulating reflection and diffraction across various materials, resulting in higher-quality reverberated audio.

On the SonicSet dataset, we conducted extensive quantitative experiments on 11 speech separation methods and 9 speech enhancement methods, thoroughly analyzing the performance of each approach. To assess the gap between SonicSet and real-world environments, we created a speech separation dataset by recording raw, non-reverberant audio from the SonicSet validation set in real-life scenarios across 10 scenes, totaling 5 hours. For the speech enhancement task, we utilized the Real-MAN test set (Yang et al., 2024), which includes real-world recordings of moving sound sources, to evaluate discrepancies between the synthetic dataset and real environments. The experimental results demonstrated that models trained on SonicSet generalized well to real-world conditions, confirming the effectiveness and potential of SonicSim for speech research.

## 2    RELATED WORKS

In this section, we systematically compare the SonicSim toolkit and the generated SonicSet dataset with existing real-world datasets, synthetic datasets, and simulation toolkits (see Table 1).

**Real-world datasets:** Currently, there are very few speech separation and enhancement datasets recorded in real-world environments (Yang et al., 2024; Watanabe et al., 2020). This is primarily because most speech datasets are designed for speech recognition evaluation and only provide transcription annotations, lacking the ground-truth labels needed for supervised learning tasks such as speech separation and enhancement. Recently, the RealMAN dataset introduced real-world microphone array recordings of speech and noise (Yang et al., 2024). Although the gap between the RealMAN dataset and real-world application scenarios is relatively small, its high annotation cost and limited scalability hinder model adaptation to diverse scenarios. Our proposed SonicSim addresses these limitations by offering a highly customizable and realistic synthetic data generation method, filling the gap in scale and scene diversity present in existing datasets.

| Datasets | Geometry | Occlusion | Material | Scalability | Cost | Tools | Src Type | Tasks |
|---|---|---|---|---|---|---|---|---|
| WHAMR! 2020 | Cuboid | ✗ | ✗ | ✓ | Low | ✓ | Static | SS/SE |
| LibriCSS 2020b | Cuboid | ✓ | ✓ | ✗ | High | ✗ | Static | SS |
| DNS Challenge 2020 | Cuboid | ✗ | ✗ | ✓ | Low | ✓ | Static | SE |
| Chime6 2020 | Variable | ✓ | ✓ | ✗ | High | ✗ | Static | SS |
| LRS2 2023 | Variable | ✓ | ✓ | ✗ | High | ✗ | Static | SS |
| RealMan 2024 | Variable | ✓ | ✓ | ✗ | High | ✗ | Dynamic | SE |
| *SonicSet (ours)* | Variable | ✓ | ✓ | ✓ | Low | ✓ | Dynamic | SS/SE |

Table 1: Comparison of different datasets for speech separation and enhancement. "Geometry" refers to the room geometry. "Occlusion" indicates whether objects obstruct the sound source and microphone. "Material" shows whether the material properties of objects are considered. "Scalability" refers to the ability to increase the amount of data. "Cost" represents the difficulty of simulating or collecting data. "Tools" indicates whether there are toolkits for data collection. "Src Type" indicates the source type. "SS" and "SE" denote the speech separation and enhancement tasks.

**Synthetic datasets:** Synthetic datasets (Reddy et al., 2020; Cosentino et al., 2020; Maciejewski et al., 2020; Chen et al., 2020b) generate data by convolving Room Impulse Responses (RIRs) with multiple source or noise signals to enhance realism, often using image source methods (Allen & Berkley, 1979). While datasets like WHAMR! (Maciejewski et al., 2020) introduce synthetic reverberation and noise, they are created under controlled conditions, limiting their authenticity and inability to fully reflect real-world performance of speech separation and enhancement methods. In contrast, SonicSet ensures acoustic plausibility and offers a wider range of customizable options, such as varying geometric structures and multiple microphone array configurations, providing more challenging and realistic environments for these tasks.

**Reverberation simulation tools:** Recent advances in reverberation simulation tools have significantly contributed to the field. FRAM-RIR (Luo & Yu, 2023a) uses a stochastic approximation of the image source method for rapid RIR simulation. RIR-Generator (Tang et al., 2020) allows parameter settings based on the image source method, and Pyroomacoustics (Scheibler et al., 2018) offers a flexible Python library for fast simulation in 2D or 3D rooms. For complex acoustic environments,

COMSOL Multiphysics (Multiphysics, 1998) employs finite element methods for detailed room modeling. Compared to these tools, SonicSim based on Habitat-sim (Savva et al., 2019) surpasses others in acoustic realism, producing synthetic audio that closely resembles real-world recordings. It also provides practical features for efficiently creating diverse acoustic scenes tailored to specific needs—capabilities lacking in most existing reverberation simulation tools.

## 3 MOVING SOUND SOURCE SUITE

The moving sound source suite consists of two core components: the customizable data generator *SonicSim* and the moving sound source dataset *SonicSet*. SonicSim leverages existing datasets to create data tailored for speech separation and enhancement.

### 3.1 CUSTOMIZABLE DATA GENERATOR: SONICSIM

The customizable dataset generator, SonicSim, is specifically designed to generate datasets for speech tasks. Built on Habitat-sim (Savva et al., 2019), SonicSim leverages its highly realistic audio renderer (demonstrated in (Chen et al., 2022a)) and high-performance 3D simulator to produce high-quality audio data adapted to various acoustic environments. SonicSim provides a rich set of annotations, including source and microphone position maps, clean audio, and audio with reverberation and noise, without incurring additional data collection costs. More importantly, SonicSim provides users with extensive control over the dataset generation process, allowing customization of scene layouts, scene materials, source and microphone positions, and microphone types, while ensuring physical realism through its physics engine. By adjusting these configurations, SonicSim can flexibly generate diverse acoustic environment data to meet the specific requirements of various tasks. In Appendix A, we discussed in detail the impact of environmental factors on model performance. The following main functions are included in SonicSim, as shown in Figure 1.

- **3D scene import**: Through Habitat-sim (Savva et al., 2019), SonicSim can import various realistic 3D assets generated through simulations or scans, such as those from the Matterport3D dataset (Chang et al., 2017). SonicSim leverages Habitat-sim[1] to interpret metadata and structural information from external datasets and convert them into Habitat-sim's native scene format. Geometric data, material properties, and semantic annotations are transformed to ensure that the imported scenes maintain their original high fidelity. Importing 3D scenes with various shapes, such as cylinders, enables Habitat-sim to simulate different acoustic properties according to different spatial layouts. SonicSim allows users to add additional objects to the scene; however, the 3D scenes themselves cannot be modified by users, as they are fixed during the import process.

- **Acoustic environment simulation:** SonicSim uses Habitat-sim to simulate various acoustic features in 3D environments[2]. Its capabilities include: 1) accurately simulating sound reflections within room geometries using indoor acoustic modeling and bidirectional path tracing algorithms. This effectively simulates the acoustic effect of different objects blocking sound.; 2) mapping the semantic labels of 3D scenes to material properties to set the acoustic features of different surfaces, such as absorption, scattering, and transmission coefficients. SonicSim supports a wide range of materials and objects, and users can modify the physical and categorical attributes of materials; 3) extended Habitat-sim acoustic simulation capabilities enable the synthesis of moving sound source data based on sound source paths.

- **Microphone types:** Habitat-sim offers comprehensive microphone configurations, supporting various audio formats such as mono, binaural, and ambisonics[3]. Additionally, we have integrated common linear and circular microphone arrays, allowing users to customize the shape of microphone arrays to meet different experimental requirements. Specifically, we first define multiple single-channel receivers corresponding to the number of microphones in the array. Next, we configure the spatial relative coordinates of these single-channel receivers based on the desired array shape. Finally, we bind these receivers together and adjust the absolute coordinates to set the position of the microphone array within the environment. We provided a flexible API that enables researchers to represent the desired array layout by inputting custom functions.

---

[1] https://aihabitat.org/docs/habitat-sim/habitat_sim.sim.SimulatorConfiguration.html

[2] https://aihabitat.org/docs/habitat-sim/habitat_sim.sensor.AudioSensor.html

[3] https://github.com/facebookresearch/habitat-sim/blob/main/docs/AUDIO.md

- **Source and microphone positions:** SonicSim allows users to customize or randomly set the positions of sound sources and microphones, which can be defined within the global coordinate system of the environment. Apart from static positioning, SonicSim also supports generating motion trajectories for moving sound sources and microphones based on specified start and endpoints. This feature is implemented by interpolating between defined points using navigable paths and leveraging Habitat-sim's physics engine to update the positions of moving entities over time. As the sound sources and microphones move along their trajectories, the system dynamically calculates the acoustic response in real-time. SonicSim accounts for dynamic changes in distance, occlusion, and environmental interactions, continuously updating the sound propagation paths to simulate evolving reverberation. By integrating these capabilities, SonicSim provides a highly realistic simulation environment for scenarios involving moving sound sources.

## 3.2 MOVING SOUND SOURCE DATASET: SONICSET

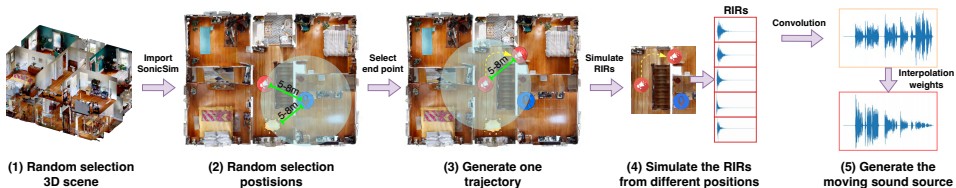

Figure 3: An automatic simulation pipeline for moving sound sources. The positions of the sound sources, noises and microphones are all randomly generated. The five RIRs shown in the figure are just for demonstration purposes; the actual process involves more RIRs.

We used SonicSim to construct a moving sound source dataset, SonicSet, specifically designed for research on moving speech separation and enhancement tasks. This dataset supports diverse acoustic scenarios by simulating microphones, sound sources, and noise sources randomly placed in the environment. A detailed data analysis can be found in Appendix B. As shown in Figure 3, the entire data generation process is outlined, ensuring the validity of the sound source and microphone configurations under varying positions and dynamic changes.

### 3.2.1 DATA SOURCE

The data we used consists of two main components: 3D assets and audio data. For the 3D assets, we utilized the Matterport3D dataset (Chang et al., 2017), a large-scale indoor RGB-D dataset containing 90 building-level scenes, representing a wide range of real-world environments. The semantic segmentation annotations from Matterport3D were used to assign acoustic material properties, enabling the creation of more realistic acoustic environments. In SonicSet, the training set includes 62 scenes, the validation set contains 19 scenes, and the test set includes 9 scenes.

For the audio data, we used speech from the LibriSpeech dataset (Panayotov et al., 2015) and noise from FSD50K (Fonseca et al., 2021) and FMA (Defferrard et al., 2017). LibriSpeech consists of approximately 1,000 hours of 16 kHz read speech from audiobooks; we selected *train-clean-360* for training, *dev-clean* for validation, and *test-clean* for testing. FSD50K, containing 51,197 manually labeled sound events totaling over 100 hours and covering 200 categories, was downsampled to 16 kHz using Librosa (McFee et al., 2015) and split into training, validation, and test sets in a 7:2:1 ratio. For FMA, an open music dataset, we used a pre-trained BSRNN model (Luo & Yu, 2023b) to remove vocals, ensuring speech separation was unaffected during data synthesis. We then downsampled the music to 16 kHz and divided it similarly into training, validation, and test sets.

### 3.2.2 DATASET CONSTRUCTION

The audio simulation pipeline in the SonicSet dataset, illustrated in Figure 3, consists of the following steps. First, we selected a 3D scene from the Matterport3D dataset and imported it into SonicSim to initialize the acoustic environment. Second, we randomly chose positions for the microphone and the sound source within a certain layer, with the sound source's starting position and the noise source's position within a 1–8 meter radius from the microphone. Third, based on the sound source's initial position, we selected an endpoint within a 1–8 meter range from both the

microphone and the initial position, using SonicSim's trajectory function[4] to generate a movement path. Fourth, SonicSim computed the RIRs along the trajectory and convolved them with the source audio. Let $\mathbf{s}(t)$ denote the source audio signal of duration $T$, and let $\mathbf{h}_j^c(t)$ represent the RIRs at discrete positions $j = 1, 2, \ldots, N$, where $c$ indexes the audio channels. The convolution at each position is $\mathbf{y}_j^c(t) = \mathbf{s}(t) * \mathbf{h}_j^c(t)$. To achieve realistic spatial movement, we interpolated the convolution results of adjacent positions using a time-varying interpolation weight $\alpha(t) \in [0, 1]$, calculated based on the Euclidean distances between the current position of the moving source $\mathbf{r}_t$ and the adjacent positions: $\alpha(t) = \frac{\text{dist}(\mathbf{r}_{j+1}, \mathbf{r}_t)}{\text{dist}(\mathbf{r}_j, \mathbf{r}_{j+1})}$, where $\text{dist}(\mathbf{r}_a, \mathbf{r}_b)$ denotes the Euclidean distance between positions $\mathbf{r}_a$ and $\mathbf{r}_b$. This ensures $\alpha(t)$ transitions smoothly from 0 to 1 as the source moves from $\mathbf{r}_j$ to $\mathbf{r}_{j+1}$. The interpolated audio signal is then computed as $\mathbf{y}(t) = (1 - \alpha(t)) \cdot \mathbf{y}_j^c(t) + \alpha(t) \cdot \mathbf{y}_{j+1}^c(t)$, dynamic acoustic changes as the source moves (Slaney et al., 1996; Freeland et al., 2002). Finally, interpolated audio segments were concatenated to form a moving source. SonicSim adjusts the source's movement speed based on trajectory length to ensure compatibility with the fixed duration $T$.

To create realistic synthetic mixed audio, we set each mixed audio clip to 60 seconds. Speech segments consisted of 3-5 full audio clips from the same LibriSpeech speaker, arranged into a 60-second sequence with random start positions within 0-8 seconds to ensure varied overlap rates, utilizing SonicSim for moving sound source synthesis. For noise data, we randomly selected 6-8 environmental noise segments from FSD50K and 6-8 musical noise segments from FMA, arranging them into 60-second clips with random start positions within 0-4 seconds. Volume levels were adjusted using Loudness Units relative to Full Scale (LUFS): speech at -17 LUFS, environmental noise at -21 LUFS, and musical noise at -24 LUFS. Mixed audio was generated by randomly choosing different speakers and adding environmental or musical noise, with background and musical noise samples including diverse, time-varying content. In each dataset group (three speech clips plus one environmental noise clip and one musical noise clip), simulations were conducted within the same scene, simulating speech as moving sources and noise as static sources. An example of SonicSet is presented in Figure 5, with corresponding audio metadata and generation hyperparameters provided in JSON files 1 and 2 in Appendix C.

# 4 BENCHMARK I: SPEECH SEPARATION

## 4.1 PROBLEM DEFINITION

Speech separation aims to isolate individual speech signals from a mixture containing multiple speakers, as shown in Figure 4(a). This task is critical for real-world applications such as meetings and telephone conversations. A detailed explanation of the task pipeline is provided in Appendix D.1.

## 4.2 BENCHMARK MODELS

We selected several popular models that have demonstrated excellent performance in speech separation as benchmarks, including Conv-TasNet (2019), DPRNN (2020), DPTNet (2020a), SuDORM-RF (2020), A-FRCNN (2021), SKIM (2022d), TDANet (2023), BSRNN (2023b), TF-GridNet (2023), Mossformer (2023), and Mossformer2 (2024). For detailed information about the benchmark models, see Appendix D.2. The PyTorch implementations and pre-trained weights for all benchmark models are publicly available[5].

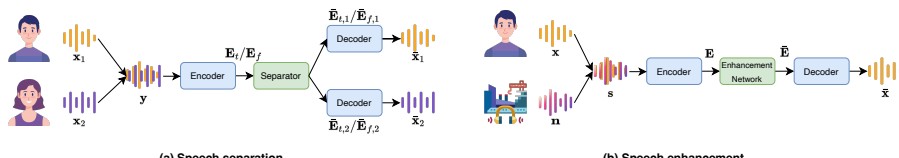

(a) Speech separation            (b) Speech enhancement

Figure 4: Overall pipeline for speech separation and enhancement.

---

[4]The trajectory is generated using Habitat-sim's API, specifically the "habitat_sim.ShortestPath" method.
[5]https://github.com/JusperLee/SonicSim/tree/main/separation

## 4.3 DATASET DETAILS

We report the performance of trained baseline models using the LRS2-2Mix (Li et al., 2023), Libri2Mix (Cosentino et al., 2020), and SonicSet datasets, which we tested on a real-world moving speech separation dataset collected by us. We used the test sets from LRS2-2Mix and Libri2Mix to test the models' generalization and transfer performance across different scenarios. Please refer to Appendix D.3 for detailed descriptions of these datasets.

## 4.4 IMPLEMENTATION DETAILS

To adapt the WSJ0-2Mix model (originally for 8 kHz) to our 16 kHz dataset, we doubled the window length and step size of the encoder and decoder, keeping other hyperparameters unchanged. We trained the model using the Adam optimizer (Kingma, 2014) with a batch size of 1 on randomly sampled 3s audio clips, employing negative SNR as the loss function. The initial learning rate was set to $1 \times 10^{-3}$, with gradient clipping and early stopping strategies applied. Detailed hyperparameters and training settings are provided in Appendix D.4 and D.5. For experiments, we used SonicSet, public datasets, and real-world datasets, with each mixed audio containing two different speakers at 16 kHz. The baseline model was pre-trained on SonicSet using a dynamic augmentation strategy (Luo & Yu, 2023b): two speakers are randomly selected from three audio tracks for mixing, adding environmental or music noise as required, and a 3s segment is randomly extracted from the 60s audio for training. In SonicSet, we used the full test set and processed with a 6s inference window and a 3s sliding window. For the LRS2-2Mix (Li et al., 2023) and Libri2Mix (Cosentino et al., 2020) datasets, 2s and 3s audio clips were used for training and testing, respectively.

## 4.5 EVALUATION METRICS

We employed a series of metrics to assess the performance of the speech separation benchmark models comprehensively. These metrics cover multiple dimensions of audio quality, including signal quality (SI-SNR (Le Roux et al., 2019) and SDR (Vincent et al., 2006)), speech intelligibility (STOI (Taal et al., 2011) and WER), and subjective quality perception (PESQ (Rix et al., 2001), NISQA (Mittag et al., 2021) and SigMOS (Ristea et al., 2024) ). Details are given in Appendix D.6.

## 4.6 RESULTS AND ANALYSIS

| Method | SI-SDR ↑ | SDR ↑ | NB-PESQ ↑ | WB-PESQ ↑ | STOI (%) ↑ | MOS Overall ↑ | NISQA ↑ | WER (%) ↓ |
|---|---|---|---|---|---|---|---|---|
| Conv-TasNet | 2.18/2.45/3.02 | 3.09/3.24/4.82 | 1.98/2.03/2.12 | 1.27/1.31/1.55 | 59.73/60.33/65.32 | 2.07/2.11/2.25 | 1.23/1.26/1.47 | 98.33/87.04/74.85 |
| DPRNN | 2.23/2.81/3.71 | 2.91/3.54/4.34 | 1.92/2.05/2.18 | 1.25/1.32/1.62 | 60.02/64.76/70.13 | 1.93/2.12/2.18 | 1.15/1.21/1.36 | 91.05/72.63/55.34 |
| DPTNet | 4.76/5.53/7.42 | 5.63/6.68/8.52 | 2.12/2.32/2.68 | 1.87/1.91/2.12 | 71.83/73.42/77.73 | 2.13/2.10/2.19 | 1.44/1.50/1.68 | 51.19/48.18/38.17 |
| SuDoRM-RF | 3.44/4.79/5.85 | 4.22/5.26/6.78 | 2.08/2.18/2.41 | 1.58/1.62/1.87 | 67.77/72.38/73.39 | 2.14/2.09/2.13 | 1.33/1.42/1.51 | 65.22/55.47/48.54 |
| A-FRCNN | 3.65/4.87/6.02 | 4.38/5.67/6.80 | 2.08/2.21/2.43 | 1.65/1.68/1.90 | 69.10/70.23/73.85 | 2.22/2.21/2.24 | 1.37/1.48/1.61 | 68.27/54.32/47.93 |
| TDANet | 3.90/5.15/6.11 | 4.71/5.98/7.10 | 2.15/2.28/2.50 | 1.72/1.69/1.94 | 69.95/71.14/74.59 | 2.21/2.20/2.23 | 1.36/1.49/1.60 | 58.40/54.39/43.60 |
| SKIM | 2.31/2.87/3.33 | 2.99/3.67/4.13 | 1.97/2.03/2.07 | 1.37/1.45/1.63 | 62.11/64.42/66.67 | 2.06/1.99/2.03 | 1.17/1.20/1.29 | 77.02/70.54/53.84 |
| BSRNN | 3.68/5.09/6.15 | 4.46/5.96/6.93 | 2.10/2.22/2.59 | 1.79/1.71/2.07 | 71.26/73.22/76.06 | 2.23/2.18/2.37 | 1.37/1.48/1.67 | 57.63/53.59/48.64 |
| TF-GridNet | 6.63/8.27/11.82 | 7.52/9.10/12.59 | 2.54/2.71/3.05 | 2.09/2.28/2.40 | 79.21/80.34/85.50 | 2.27/2.35/2.54 | 1.59/1.68/1.82 | 34.64/30.21/20.50 |
| Mossformer | 5.72/7.94/10.72 | 6.54/8.78/11.63 | 2.51/2.60/2.97 | 2.18/2.23/2.31 | 75.38/79.32/81.21 | 2.10/2.19/2.34 | 1.49/1.62/1.69 | 47.33/33.84/30.44 |
| Mossformer2 | 5.87/7.81/10.57 | 6.66/8.65/11.25 | 2.56/2.58/2.98 | 2.23/2.21/2.35 | 75.50/78.94/81.00 | 2.07/2.16/2.25 | 1.50/1.60/1.71 | 42.94/33.09/29.57 |

Table 2: Comparative performance evaluation of models trained on different datasets using RealSEP with ***environmental noise***. The results are reported separately for "*trained on LRS2-2Mix*", "*trained on Libri2Mix*" and "*trained on SonicSet*", distinguished by a slash. The relative length is indicated below the value by horizontal bars.

**Comparison on real-world datasets**. To validate the acoustic simulation gap between synthetic datasets and real data, we constructed a real-world moving sound source dataset consisting of 5 hours of recorded audio (details in Appendix D.3), namely *RealSEP*, encompassing various complex acoustic environments and different types of background noise (environmental and musical noise). In Tables 2 and 3, we trained models using the LRS2-2Mix and Libri2Mix datasets to compare their performance with models trained on the SonicSet dataset. The LRS2-2Mix dataset (Li et al., 2023) contains real-world noise and reverberation. The Libri2Mix dataset (Cosentino et al., 2020) is currently the largest synthetic speech separation dataset, and its data scale is closest to SonicSet among public speech separation datasets. We trained different models on these datasets and tested them on the RealSEP datasets. In addition, we supplemented the experimental results by recording audio data of direct conversations between individuals (HumanSEP, details in Appendix D.3). We

| Method | SI-SDR ↑ | SDR ↑ | NB-PESQ ↑ | WB-PESQ ↑ | STOI (%) ↑ | MOS Overall ↑ | NISQA ↑ | WER (%) ↓ |
|---|---|---|---|---|---|---|---|---|
| Conv-TasNet | 0.34/0.48/1.13 | 0.71/0.81/2.61 | 1.63/1.63/1.90 | 1.19/1.24/1.36 | 52.79/53.61/59.40 | 1.65/1.80/1.90 | 1.19/1.22/1.44 | 97.46/94.34/75.02 |
| DPRNN | 0.43/0.52/0.56 | 0.86/0.99/1.22 | 1.65/1.64/1.75 | 1.25/1.30/1.30 | 54.83/55.68/57.40 | 1.82/1.92/2.17 | 1.10/1.14/1.16 | 94.91/85.76/61.66 |
| DPTNet | 4.56/5.31/8.90 | 5.37/6.14/9.77 | 2.24/2.17/2.41 | 1.81/1.88/1.76 | 70.62/71.71/78.06 | 2.08/2.12/2.16 | 1.33/1.42/1.59 | 51.05/46.13/36.11 |
| SuDoRM-RF | 3.10/3.35/5.57 | 3.87/4.42/6.32 | 2.03/2.00/2.10 | 1.47/1.53/1.53 | 65.19/66.21/76.22 | 2.02/2.10/2.15 | 1.28/1.35/1.43 | 68.22/61.64/50.23 |
| A-FRCNN | 3.17/3.69/5.57 | 3.96/4.52/6.29 | 2.09/2.06/2.13 | 1.51/1.57/1.54 | 66.24/67.27/77.06 | 2.13/2.14/2.22 | 1.26/1.34/1.52 | 65.79/59.45/50.41 |
| TDANet | 3.21/3.81/5.59 | 3.99/4.56/6.25 | 2.11/2.07/2.14 | 1.51/1.57/1.54 | 66.32/67.35/77.00 | 2.16/2.19/2.27 | 1.28/1.38/1.47 | 62.28/56.28/48.93 |
| SKIM | 1.92/2.27/3.09 | 2.58/2.95/3.85 | 1.81/1.87/1.89 | 1.33/1.38/1.39 | 59.73/60.66/70.58 | 1.91/2.03/2.08 | 1.18/1.20/1.24 | 77.73/70.24/58.24 |
| BSRNN | 3.37/3.98/7.66 | 4.15/4.74/8.27 | 2.11/2.10/2.37 | 1.61/1.67/1.63 | 68.44/69.50/76.43 | 2.13/2.19/2.44 | 1.31/1.37/1.60 | 58.12/52.52/46.52 |
| TF-GridNet | 6.01/8.64/10.27 | 6.98/9.97/11.41 | 2.26/2.47/2.85 | 1.55/1.90/2.02 | 78.55/79.77/81.98 | 2.27/2.39/2.49 | 1.54/1.67/1.78 | 46.23/39.71/38.04 |
| Mossformer | 4.10/6.58/8.84 | 5.06/7.79/9.80 | 2.23/2.32/2.55 | 1.56/1.83/1.89 | 71.83/72.95/80.47 | 2.20/2.26/2.36 | 1.39/1.55/1.64 | 53.98/48.78/40.81 |
| Mossformer2 | 4.18/6.92/8.64 | 5.02/7.73/9.46 | 2.28/2.30/2.48 | 1.55/1.82/1.85 | 69.61/70.69/77.57 | 2.15/2.24/2.37 | 1.39/1.50/1.53 | 58.44/52.81/47.33 |

Table 3: Comparative performance evaluation of models trained on different datasets using RealSEP with ***musical noise***. The results are reported separately for "*trained on LRS2-2Mix*", "*trained on Libri2Mix*" and "*trained on SonicSet*", distinguished by a slash.

| Method | SI-SNR ↑ | SDR ↑ | NB-PESQ ↑ | WB-PESQ ↑ | STOI (%) ↑ | MOS Ovrl ↑ | NISQA ↑ | WER (%) ↓ |
|---|---|---|---|---|---|---|---|---|
| Conv-TasNet | 4.81/4.12 | 7.13/5.38 | 2.00/1.84 | 1.46/1.42 | 73.14/63.21 | 2.10/1.81 | 1.64/1.71 | 53.82/63.21 |
| DPRNN | 4.87/4.37 | 6.65/5.73 | 2.17/1.98 | 1.63/1.50 | 77.63/73.73 | 2.11/2.07 | 1.35/1.42 | 47.81/51.33 |
| DPTNet | 11.51/11.69 | 13.00/12.80 | 2.82/2.67 | 2.35/2.13 | 87.62/84.23 | 2.32/2.23 | 1.72/1.84 | 28.13/29.05 |
| SuDoRM-RF | 8.01/6.84 | 9.70/8.34 | 2.47/2.15 | 1.98/1.66 | 81.28/77.75 | 2.25/2.12 | 1.49/1.61 | 35.61/41.37 |
| A-FRCNN | 9.17/7.59 | 10.63/9.32 | 2.70/2.52 | 2.16/2.00 | 84.82/82.14 | 2.32/2.29 | 1.65/1.76 | 35.44/33.82 |
| TDANet | 9.27/7.00 | 11.00/8.68 | 2.72/2.26 | 2.22/1.71 | 85.90/79.12 | 2.36/2.19 | 1.64/1.76 | 30.46/37.16 |
| SKIM | 7.23/6.00 | 8.78/7.42 | 2.34/2.23 | 1.86/1.75 | 79.36/77.61 | 2.11/2.10 | 1.33/1.46 | 38.92/42.82 |
| BSRNN | 9.10/6.96 | 10.86/8.66 | 2.82/2.36 | 2.26/1.76 | 85.27/79.12 | 2.45/2.32 | 1.83/1.90 | 29.86/41.73 |
| TF-GridNet | 15.38/14.37 | 16.81/15.69 | 3.58/3.45 | 3.08/2.84 | 93.32/91.80 | 2.49/2.58 | 1.91/2.05 | 12.04/14.43 |
| Mossformer | 14.72/11.80 | 15.97/13.17 | 3.02/2.82 | 2.67/2.26 | 91.13/86.15 | 2.39/2.25 | 1.86/1.98 | 21.10/26.64 |
| Mossformer2 | 14.84/11.12 | 16.09/12.34 | 3.17/2.62 | 2.83/2.09 | 91.79/83.24 | 2.40/2.20 | 1.89/2.02 | 19.51/32.65 |

Table 4: Comparison of existing speech separation methods on the SonicSet dataset. The performance of each model is listed separately for results under "*environmental noise*" and "*musical noise*", distinguished by a slash.

then evaluated the NISQA and WER results of the speech separation model trained on different data sets, as shown in Table 10 in the Appendix. The results demonstrated that models trained on the SonicSet dataset achieved overall the best separation performance on the RealSEP and HumanSEP datasets. This finding indicates that the SonicSet dataset excels in simulating more realistic acoustic scenes, making it effective for model training and evaluation.

**Comparison on the SonicSet dataset**. The results of the speech separation are shown in Table 4. Different models exhibit varying performance under environmental and musical noise conditions. Among the RNN-based models, DPRNN and BSRNN performed relatively poorly when handling musical noise. In contrast, CNN-based models (SuDoRM-RF and A-FRCNN) demonstrated more balanced performance across both environmental and musical noise scenarios. Among Transformer-based models, Mossformer2 showed the SOTA performance, especially in handling complex noise. This superiority can be attributed to incorporating the multi-head attention mechanism, which effectively captures long-term dependencies and complex frequency variations (Vaswani et al., 2017). The detailed discussion is in Appendix D.7.

# 5 BENCHMARK II: SPEECH ENHANCEMENT

## 5.1 PROBLEM DEFINITION

The speech enhancement task aims to extract high-quality target speech from a noisy signal, reducing or eliminating background noise, as shown in Figure 4(b). This task is crucial in applications such as speech recognition, speech communication, and hearing aids. A detailed explanation of this task pipeline is available in Appendix E.1.

## 5.2 BENCHMARK MODELS

In speech enhancement experiments, we selected several models: DCCRN (2020), Fullband (2021), FullSubNet (2021), Fast-FullSubNet (2022), FullSubNet+ (2022b), TaylorSENet (2022a), GaGNet (2022c), G2Net (2022b) and Inter-SubNet (2023). The details of these models are provided in Appendix E.2.

### 5.3 IMPLEMENTATION DETAILS

The hyperparameter settings for all baseline models remain consistent with the original papers. We trained the models using the Adam optimizer (Kingma, 2014) with an initial learning rate of 0.001. If the validation loss does not decrease for five consecutive epochs, we reduce the learning rate by a factor of 0.5. All audio samples are processed at a 16 kHz sampling rate. Training employs early stopping and halting if the validation loss doesn't improve for 10 epochs. During testing, we use the same inference strategy as in the speech separation task to handle long audio samples. For the speech enhancement experiments, we trained models on the VoiceBank-DEMAND (Valentini-Botinhao et al., 2016), DNS Challenge (Reddy et al., 2020), and SonicSet datasets using single-channel speech signals sampled at 16 kHz. Baseline models were pre-trained on SonicSet with the same dynamic enhancement strategy as in separation model training, extracting 3s segments from 60s audio for training. The training objective is detailed in Appendix E.3. For SonicSet, we used the full test set with a 6s inference window and a 3s sliding window to process long audio sequences. On the VoiceBank-DEMAND and DNS Challenge datasets, 2s and 3s audio clips were used for training and testing, respectively.

### 5.4 EVALUATION METRICS

In the speech enhancement task, we use a series of evaluation metrics to comprehensively evaluate the performance of the speech separation baseline model, including signal quality (SI-SNR (Le Roux et al., 2019) and SDR (Vincent et al., 2006)), speech intelligibility (STOI (Taal et al., 2011) and CER), and subjective quality perception (PESQ (Rix et al., 2001), NISQA (Mittag et al., 2021), DNSMOS (Reddy et al., 2021) and SigMOS (Ristea et al., 2024)). Since the real-world dataset, RealMAN is a Chinese dataset, we use CER as the evaluation metric for speech recognition. For these metrics, except for CER, the higher the value, the better, while for CER, the lower the value, the better. From an application perspective, CER is the most important metric, as speech enhancement is usually a pre-processing step for speech recognition. The PyTorch implementations and pre-trained weights for all baseline models have been made publicly available[6].

### 5.5 DATASET DETAILS

Unlike the speech separation task, in the speech enhancement task, we select the audio of a single speaker from the training set as the ground truth label for each iteration, mixing it with various types of noise using the same mixing method as in the speech separation task. We trained baseline models using the training sets of VoiceBank-DEMAND (Valentini-Botinhao et al., 2016), the DNS Challenge (Reddy et al., 2020) and SonicSet, and tested them on the test set of the real-world dataset (RealMAN (Yang et al., 2024)) to evaluate the effectiveness of different datasets in simulating real acoustic environments. To assess the generalization ability of models trained on SonicSet, we also tested the baseline models on VoiceBank-DEMAND and the DNS Challenge test sets.

### 5.6 RESULTS AND ANALYSIS

**Comparison on real-recorded dataset**. We utilized the real-world recorded moving source dataset, RealMAN (test set), to evaluate trained speech enhancement models on various speech enhancement datasets, including VoiceBank-DEMAND, DNS Challenge, and SonicSet. VoiceBank-DEMAND is a smaller dataset, while the DNS Challenge dataset comprises approximately 700 hours of data, with a data scale comparable to that of SonicSet. We employed the DNSMOS (Reddy et al., 2021) tool to calculate subjective metrics, while Character Error Rate (CER) results were evaluated using the same speech recognition model[7] as used with RealMAN. The results presented in Table 5 indicated that models trained on the SonicSet dataset achieved overall the best results across multiple metrics. Please note that for most practical applications, CER is the most important metric. Moreover, we supplemented the experimental results by constructing a dataset called HumanENH (details in Appendix E.4) by recording human speech audio and noise data. We then evaluated the NISQA and WER results of the speech enhancement models trained on different datasets, as shown in Table 11 in the appendix.

---

[6] https://anonymous.4open.science/r/SonicSim-ICLR2025/enhancement
[7] https://huggingface.co/espnet/pengcheng_guo_wenetspeech_asr_train_asr_raw_zh_char

Based on these findings, we infer that the synthetic dataset SonicSet effectively simulates the acoustic environments of real moving source scenarios. This provides significant flexibility and cost-effectiveness for constructing large-scale datasets, positioning synthetic datasets as an effective alternative for addressing complex environments.

| Method | SDR ↑ | WB-PESQ ↑ | MOS Sig ↑ | MOS Bak ↑ | MOS Overall ↑ | NISQA ↑ | CER (%) ↓ |
|---|---|---|---|---|---|---|---|
| DCCRN | −1.10/1.87/1.95 | 1.11/1.24/1.26 | 2.26/3.25/2.44 | 2.90/2.12/3.36 | 1.90/2.19/2.27 | 1.84/1.97/1.62 | 50.65/37.56/21.70 |
| Fullband | −1.55/1.18/1.37 | 1.04/1.07/1.27 | 2.50/2.84/2.53 | 2.22/2.61/3.47 | 2.09/2.19/2.46 | 2.03/2.12/1.93 | 51.67/39.71/20.94 |
| FullSubNet | −0.75/1.48/1.92 | 1.10/1.19/1.30 | 2.40/2.73/2.69 | 2.94/2.76/3.48 | 1.94/2.24/2.46 | 2.14/2.19/1.98 | 49.23/32.39/19.15 |
| Fast-FullSubNet | −1.55/1.38/1.37 | 1.08/1.15/1.31 | 2.45/3.13/2.67 | 2.09/2.09/3.48 | 2.04/1.97/2.59 | 2.16/2.28/2.05 | 49.97/40.17/20.08 |
| FullSubNet+ | −0.58/1.64/1.32 | 1.11/1.27/1.28 | 2.44/2.51/2.59 | 2.09/2.87/3.52 | 2.07/2.31/2.46 | 2.13/2.17/1.94 | 45.22/23.98/20.48 |
| TaylorSENet | 1.06/1.78/2.26 | 1.21/1.33/1.31 | 2.44/2.68/2.47 | 2.09/2.63/2.43 | 2.10/2.23/2.33 | 1.46/1.68/0.77 | 42.55/28.19/20.64 |
| GaGNet | −0.13/1.65/2.10 | 1.07/1.27/1.30 | 2.62/2.53/2.46 | 2.44/3.16/2.41 | 2.32/2.35/2.40 | 1.39/1.61/0.77 | 44.39/34.77/21.09 |
| G2Net | 0.01/1.52/2.10 | 1.10/1.21/1.29 | 2.76/2.75/2.45 | 2.21/2.53/2.41 | 2.07/2.17/2.41 | 1.41/1.72/0.81 | 55.12/42.98/21.67 |
| Inter-SubNet | −1.62/1.35/1.61 | 1.09/1.29/1.34 | 2.13/2.67/2.70 | 3.83/2.88/3.47 | 1.83/2.40/2.51 | 2.12/2.23/1.94 | 47.73/22.96/18.73 |

Table 5: Comparative performance evaluation of models trained on different datasets using the Real-MAN dataset. The results are reported separately for "*trained on VoiceBank-DEMAND*", "*trained on DNS Challenge*" and "*trained on SonicSet*", distinguished by a slash. These MOS metrics were calculated using DNSMOS.

| Method | SI-SNR ↑ | SDR ↑ | WB-PESQ ↑ | STOI (%) ↑ | WER (%) ↓ | MOS Noise ↑ | MOS Rvrb ↑ | MOS Sig ↑ | MOS Ovrl ↑ | NISQA ↑ |
|---|---|---|---|---|---|---|---|---|---|---|
| DCCRN | 8.41/11.56 | 11.29/11.98 | 2.17/2.00 | 87.39/85.04 | 21.78/25.13 | 2.94/3.30 | 3.01/3.51 | 2.80/2.94 | 2.39/2.59 | 2.01/1.98 |
| Fullband | 7.82/10.07 | 8.34/11.10 | 2.34/2.02 | 89.67/86.99 | 22.04/25.27 | 3.30/3.13 | 3.04/2.99 | 2.95/2.88 | 2.54/2.46 | 2.05/2.07 |
| FullSubNet | 9.48/11.60 | 11.92/12.31 | 2.48/2.22 | 90.44/88.04 | 20.01/20.82 | 3.24/3.34 | 3.08/3.05 | 2.98/3.05 | 2.54/2.63 | 2.13/2.14 |
| Fast-FullSubNet | 8.14/10.36 | 8.71/11.92 | 2.41/2.08 | 90.04/87.82 | 21.13/24.98 | 3.31/3.22 | 3.03/3.03 | 2.99/2.93 | 2.58/2.51 | 2.09/2.13 |
| FullSubNet+ | 8.93/10.64 | 11.07/12.31 | 2.35/1.99 | 89.17/86.36 | 20.73/24.11 | 3.12/3.02 | 2.97/2.93 | 2.91/2.82 | 2.47/2.38 | 2.03/2.06 |
| TaylorSENet | 10.11/12.18 | 12.67/13.14 | 2.45/2.33 | 89.38/85.53 | 21.61/23.46 | 2.72/2.76 | 2.92/2.92 | 2.65/2.65 | 2.22/2.24 | 1.61/1.66 |
| GaGNet | 10.01/12.20 | 12.78/13.13 | 2.48/2.27 | 89.57/87.47 | 21.40/23.36 | 2.77/2.78 | 2.86/2.86 | 2.64/2.64 | 2.23/2.21 | 1.62/1.64 |
| G2Net | 9.82/12.14 | 12.22/13.13 | 2.49/2.32 | 89.73/88.21 | 22.02/22.96 | 2.78/2.80 | 2.88/2.88 | 2.64/2.64 | 2.22/2.23 | 1.59/1.64 |
| Inter-SubNet | 10.34/12.07 | 12.87/13.01 | 2.61/2.28 | 91.87/88.45 | 18.83/20.07 | 3.39/3.34 | 3.10/3.11 | 3.05/3.04 | 2.62/2.64 | 2.12/2.15 |

Table 6: Comparison of speech enhancement methods using the SonicSet test set. The metrics are listed separately under "*environmental noise*" and "*musical noise*", distinguished by a slash. These MOS metrics were calculated using SigMOS.

**Comparison on the SonicSet dataset**. By comparing the performance of different speech enhancement models on the SonicSet dataset, we analyzed their enhancement effects under various types of noise. Table 6 presents the performance of the benchmark models. Sub-band splitting methods (such as FullSubNet, Fast-FullSubNet, FullSubNet+, and Inter-SubNet) performed well across multiple metrics. Their interaction processing mechanism between sub-bands significantly improved the models' ability to handle frequency band dependencies, resulting in better speech clarity and noise suppression. In contrast, although TaylorSENet and GaGNet showed relatively similar performance on SI-SNR and SDR, they fell slightly behind in the MOS series scores. The detailed discussion is in Appendix E.5.

## 6 CONCLUSIONS

We introduce a simulation tool named SonicSim, and a large-scale synthetic dataset named Sonic-Set, designed to study speech separation and enhancement tasks involving moving sound sources. By integrating the Habitat-sim platform, we developed a tool capable of simulating complex acoustic environments, supporting moving sound sources and multi-scene audio generation. Baseline experiments demonstrate that models pre-trained on the SonicSet dataset exhibit strong generalization abilities across various public benchmark datasets and real-world recorded datasets, effectively narrowing the gap between simulation and real-world scenarios. Through continuous improvements in the simulation tool and optimization of model algorithms, future work could further advance the deployment of speech tasks in complex environments.

**Limitation**: The realism of SonicSim's audio simulation is constrained by the level of detail in 3D scene modeling. When there are gaps or incomplete structures in the imported 3D scenes, the system cannot accurately simulate the reverberation effects in the current environment.

ACKNOWLEDGMENTS

This work was supported in part by the National Key Research and Development Program of China (No. 2021ZD0200301) and the National Natural Science Foundation of China (No. U2341228).

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

## A ENVIRONMENTAL FACTOR

We randomly selected 300 audio files from the Librispeech (Panayotov et al., 2015) test-clean set. 200 of these were used to generate noisy mixtures containing two speakers, and the other 100 were used to generate noisy audio from a single speaker. The same audio files were used in both cases. To fix the scene, we chose a cuboid room with no obstructions but included room materials and fixed the positions of the microphone, sound source, and noise. In the experiment on occlusion, we used Habitat-sim's API "add_object" to place objects between the microphone and the sound source to simulate the occlusion in sound wave propagation. When studying the effect of room shape, we chose a cylindrical room from the Matterport3D dataset (Chang et al., 2017) and fixed the relative positions of the microphone, sound source, and noise. In addition, we completely removed the root material information from the experiment to explore the effect of the material.

We selected the two methods with the best speech separation and enhancement performance and verified the impact of different environmental factors on the model performance. The experimental results are shown in Tables 7 and 8. The results show that material has the most significant effect on model performance, while the impact of room shape is relatively small. However, all these factors have a certain degree of interference in model performance. Therefore, a simulated dataset using these environmental factors is generated to train the model and improve its generalization ability.

| Methods | with/without obstacles | | cylinder/cuboid | | with/without materials | |
|---|---|---|---|---|---|---|
| | SDR ($\uparrow$) | WER (%) ($\downarrow$) | SDR ($\uparrow$) | WER (%) ($\downarrow$) | SDR ($\uparrow$) | WER (%) ($\downarrow$) |
| Mossformer | 10.70/11.84 | 29.90/24.31 | 11.18/11.84 | 27.17/24.31 | 11.84/10.12 | 24.31/31.10 |
| Mossformer2 | 10.41/11.60 | 31.89/25.86 | 10.78/11.60 | 30.54/25.86 | 11.60/9.87 | 25.86/33.58 |

Table 7: Comparison of the performance of speech separation models for different environmental factors.

| Methods | with/without obstacles | | cylinder/cuboid | | with/without materials | |
|---|---|---|---|---|---|---|
| | SDR ($\uparrow$) | WER (%) ($\downarrow$) | SDR ($\uparrow$) | WER (%) ($\downarrow$) | SDR ($\uparrow$) | WER (%) ($\downarrow$) |
| FullSubNet | 9.67/11.05 | 25.76/19.80 | 10.56/11.05 | 21.94/19.80 | 11.05/10.23 | 19.80/21.17 |
| Inter-SubNet | 10.78/12.69 | 22.96/17.68 | 11.50/12.69 | 19.30/17.68 | 12.69/11.26 | 17.68/19.88 |

Table 8: Comparison of the performance of speech enhancement models for different environmental factors.

## B DATASET ANALYSIS

The dataset statistics are shown in Table 9. SonicSet dataset offers ground-truth labels for speech separation and enhancement for supervised learning, which captures different audio samples for data synthesis in the same scene, with acoustic environments that closely resemble real-world conditions.

SonicSet contains 57,596 speech moving trajectories across 90 scenes. The training set includes 57,102 trajectories, while the validation and test sets each contain 247 trajectories, covering most possible positions within indoor scenes. The dataset comprises 1,001 speakers, with 921 speakers in the training set, 40 in the validation set, and 40 in the test set, ensuring that no speaker appears in different subsets, which maintains speaker-independent training. Through our data construction process, SonicSet generates approximately 952 hours of training data, 4 hours of validation data,

and 4 hours of test data. SonicSet provides researchers with a high-quality and complex moving sound source dataset for speech separation and enhancement methods.

| Datasets | Speakers | Utterances | Duration (h) | Noise | Reverb | Dynamic |
|---|---|---|---|---|---|---|
| **Speech enhancement** | | | | | | |
| TIMIT (1990) | 630 | 6,300 | 5 | ✓ | ✗ | ✗ |
| VoiceBank-DEMAND (2016) | 110 | 400 | 44 | ✓ | ✗ | ✗ |
| DNS Challenge (2020) | ˜11k | ˜100k | 760 | ✓ | ✓ | ✗ |
| RealMan (2024) | 55 | - | 81 | ✓ | ✓ | ✓ |
| **Speech separation** | | | | | | |
| WSJ0 (2016) | 191 | 28,000 | 43 | ✗ | ✗ | ✗ |
| Libri2Mix (2020) | 1001 | 56,800 | 232 | ✗ | ✗ | ✗ |
| LibriCSS (2020b) | 40 | ˜1000 | 10 | ✓ | ✓ | ✗ |
| LRS2-2Mix (2023) | 100 | 48,164 | 50 | ✓ | ✓ | ✗ |
| **Speech separation and enhancement** | | | | | | |
| WHAM! (2019) | 191 | 28,000 | 43 | ✓ | ✗ | ✗ |
| WHAMR! (2020) | 191 | 28,000 | 43 | ✓ | ✓ | ✗ |
| *SonicSet (ours)* | 1001 | 57,596 | 960 | ✓ | ✓ | ✓ |

Table 9: Comparison of speech datasets and their characteristics. "Speakers" refer to the number of unique speakers. "Utterances" stands for the total number of utterances, and "Duration" represents the total duration of the dataset in hours. "Noise", "Reverb", and "Dynamic" indicate whether the dataset contains noisy conditions, reverberation, and dynamic acoustic environments, respectively.

## C    SONICSET DEMO

We provide a detailed description of the composition and structure of the SonicSet dataset. Each data group consists of audio files from three mobile sound sources, two types of noise, and trajectory information regarding the movement of these sources. Specifically, each sound source audio file comprises multiple audio segments in FLAC format, with each segment associated with precise start and end times along with corresponding textual content. In addition to speech data, SonicSet includes two categories of interference: background noise and musical noise. The background noise data captures various non-speech sounds in the environment, while the musical noise data consists of music audio recorded within relatively consistent time intervals. Each data group is also accompanied by detailed information on sound source trajectories, microphone locations, and noise positions, illustrating the movement patterns of sound sources in different environments. This data format, combining speech, noise, and motion information, provides valuable training data for tasks such as sound source localization (Grumiaux et al., 2022), voice activity detection (Sharma et al., 2021), and speech recognition (Malik et al., 2021). With its rich audio and trajectory information, the SonicSet dataset serves as an ideal testing platform for a variety of speech processing tasks, including noisy speech recognition, sound source separation, and source tracking, thereby enhancing the performance of speech and audio-related technologies.

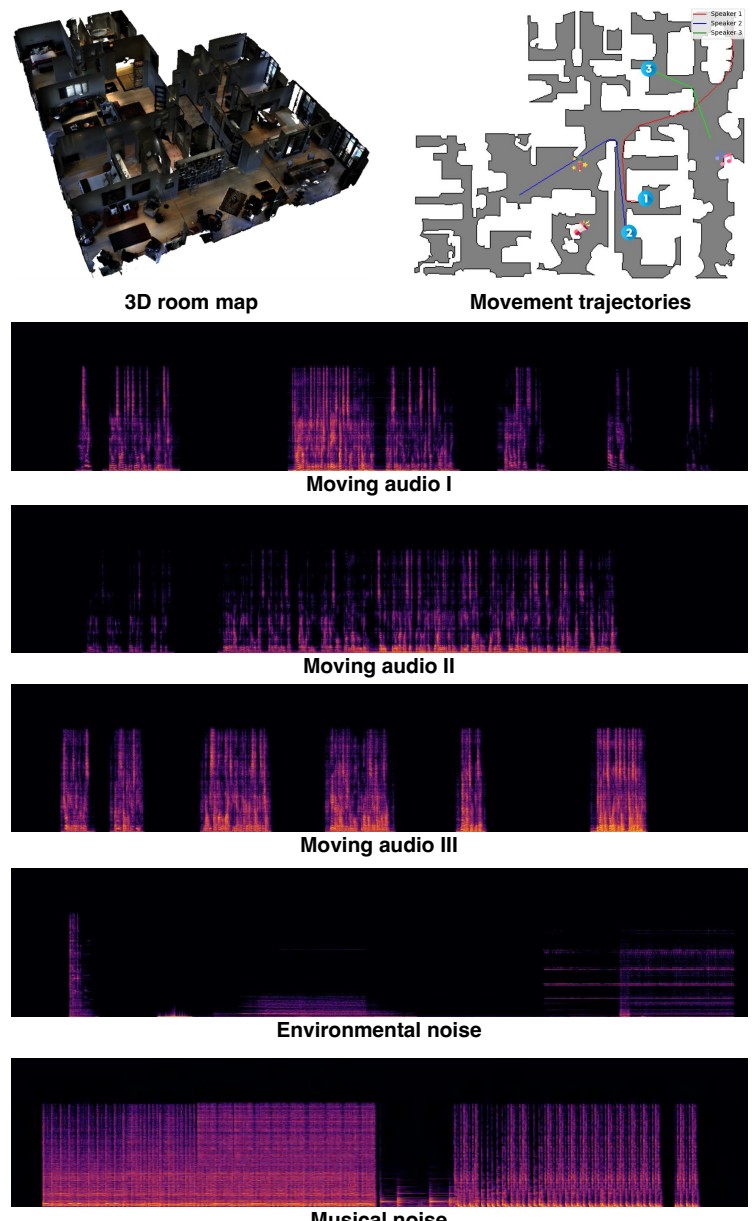

Figure 5: An example from the SonicSet data set. Each set of data includes three moving speaker audio files, two types of noise, and the trajectories of the sound source movement.

JSON file 1: The hyperparameters required for audio generation.

```
{
  "microphone": {
    "type": "monaural",  // Options: "monaural", "binaural", "Ambisonics"
        , "Custom array"
    "position": {
      "x": 0.0,
      "y": 1.5,
      "z": 0.0
    }
  },
  "sound_source": {
    "audio_file": "source_audio.wav",
```

```
      "start_point": {
        "x": 5.0,
        "y": 1.5,
        "z": -3.0
      },
      "end_point": {
        "x": -2.0,
        "y": 1.5,
        "z": 4.0
      },
      "movement_type": "dynamic",  // Options: "static", "dynamic"
      "audio_settings": {
          "duration": 10.0,          // Duration in seconds
          "sampling_rate": 44100,  // In Hz
      },
    },
  "noise_source": {
    "audio_file": "background_noise.wav",
    "position": {
        "x": 2.0,
        "y": 1.5,
        "z": -1.0
    }
  },
  "audio_settings": {
    "duration": 10.0,          // Duration in seconds
    "sampling_rate": 44100,  // In Hz
  },
  "environment": {
    "scene": "Matterport3D",
  }
}
```

JSON file 2: The raw information corresponding to the audio in Figure 5 is stored in a file in JSON format. It can be used to train voice activity detection and speech recognition.

```
{"source1": {
        "audio": [
            "672-122797-0033.flac",
            "672-122797-0070.flac",
            "672-122797-0045.flac",
            "672-122797-0054.flac",
            "672-122797-0029.flac",
            "672-122797-0053.flac"
        ],
        "start_end_points": [
            [83071, 103631],
            [118366, 218686],
            [360049, 583409],
            [632894, 700894],
            [772607, 821407],
            [871263, 918543]
        ],
        "words": [
            "A STORY",
            "THE GOLDEN STAR OF TINSEL WAS STILL ON THE TOP OF THE TREE
                AND GLITTERED IN THE SUNSHINE",
            "TIME ENOUGH HAD HE TOO FOR HIS REFLECTIONS FOR DAYS AND
                NIGHTS PASSED ON AND NOBODY CAME UP AND WHEN AT LAST
                SOMEBODY DID COME IT WAS ONLY TO PUT SOME GREAT TRUNKS IN
                A CORNER OUT OF THE WAY",
            "I KNOW NO SUCH PLACE SAID THE TREE",
            "HOW IT WILL SHINE THIS EVENING",
            "THEY WERE SO EXTREMELY CURIOUS"
```

```
        ]
    },
    "source2": {
        "audio": [
            "908-31957-0014.flac",
            "908-157963-0007.flac"
        ],
        "start_end_points": [
            [91122, 212082],
            [268394, 792714]
        ],
        "words": [
            "A RING OF AMETHYST I COULD NOT WEAR HERE PLAINER TO MY SIGHT
                THAN THAT FIRST KISS",
            "THE LILLY OF THE VALLEY BREATHING IN THE HUMBLE GRASS
                ANSWERD THE LOVELY MAID AND SAID I AM A WATRY WEED AND I
                AM VERY SMALL AND LOVE TO DWELL IN LOWLY VALES SO WEAK
                THE GILDED BUTTERFLY SCARCE PERCHES ON MY HEAD YET I AM
                VISITED FROM HEAVEN AND HE THAT SMILES ON ALL WALKS IN
                THE VALLEY AND EACH MORN OVER ME SPREADS HIS HAND SAYING
                REJOICE THOU HUMBLE GRASS THOU NEW BORN LILY FLOWER"
        ]
    },
    "source3": {
        "audio": [
            "61-70968-0021.flac",
            "61-70968-0052.flac",
            "61-70968-0030.flac",
            "61-70968-0050.flac",
            "61-70970-0006.flac",
            "61-70968-0013.flac"
        ],
        "start_end_points": [
            [63669, 106709],
            [127764, 170164],
            [245602, 336562],
            [408129, 497409],
            [578830, 616190],
            [754238, 825438]
        ],
        "words": [
            "SURELY WE CAN SUBMIT WITH GOOD GRACE",
            "BUT WHO IS THIS FELLOW PLUCKING AT YOUR SLEEVE",
            "NOW BE SILENT ON YOUR LIVES HE BEGAN BUT THE CAPTURED
                APPRENTICE SET UP AN INSTANT SHOUT",
            "HE MADE AN EFFORT TO HIDE HIS CONDITION FROM THEM ALL AND
                ROBIN FELT HIS FINGERS TIGHTEN UPON HIS ARM",
            "NEVER THAT SIR HE HAD SAID",
            "BEFORE THEM FLED THE STROLLER AND HIS THREE SONS CAPLESS AND
                TERRIFIED"
        ]
    },
    "noise": {
        "audio": [],
        "start_end_points": [
            [26850, 940228]
        ]
    },
    "music": {
        "audio": [],
        "start_end_points": [
            [13482, 918705]
        ]
    }
}
```

# D    BENCHMARK I: SPEECH SEPARATION

## D.1    PROBLEM DEFINITION

Given an input signal $\boldsymbol{y} = \sum_{c=0}^{C} \boldsymbol{x}_i + \boldsymbol{n}, \boldsymbol{y} \in \mathbb{R}^{1 \times T}$ that contains audio $\boldsymbol{x}_i \in \mathbb{R}^{1 \times T}$ from $C$ speakers and noise $\boldsymbol{n} \in \mathbb{R}^{1 \times T}$, the objective of speech separation is to extract each speaker's speech $\bar{\boldsymbol{x}}_i \in \mathbb{R}^{1 \times T}$ and assign it to different output channels, where $T$ denotes the length of audio. Currently, most speech separation methods adopt an encoder-separator-decoder framework (Wang & Chen, 2018).

In time-domain approaches, the encoder converts $\boldsymbol{y}$ into high-dimensional features $\boldsymbol{E}_t \in \mathbb{R}^{N \times T_t}$ through 1D convolutional layers, where $N$ and $T_t$ denote the feature dimension and length, respectively. In time-frequency domain approaches, the encoder applies Short-Time Fourier Transform (STFT) to convert $\boldsymbol{y}$ into time-frequency features $\boldsymbol{E}_f \in \mathbb{C}^{F \times T_f}$, where $F$ and $T_f$ represent the frequency and the time dimensions, respectively. Next, $\boldsymbol{E}_t$ or $\boldsymbol{E}_f$ is passed to the separator to obtain the feature representations of individual speakers, $\bar{\boldsymbol{E}}_{t,i} \in \mathbb{R}^{N \times T_t}$ or $\bar{\boldsymbol{E}}_{f,i} \in \mathbb{C}^{F \times T_f}$. Finally, the decoder reconstructs the target waveform $\bar{\boldsymbol{x}}_i$ from $\bar{\boldsymbol{E}}_{t,i}$ or $\bar{\boldsymbol{E}}_{f,i}$ using transposed convolutional layers or inverse STFT, thereby achieving the final separated speech sources.

## D.2    BENCHMARK MODELS

**Conv-TasNet** (Luo & Mesgarani, 2019): As the first time-domain speech separation model, it employs a dilated convolutional network, surpassing traditional frequency-domain methods in separation performance. This model processes audio directly in the time domain, enhancing clarity compared to frequency-based methods.

**DPRNN** (Luo et al., 2020): A time-domain model based on bidirectional LSTM (BLSTM), it introduced a dual-path architecture for intra- and inter-block modeling, significantly improving the ability to capture long-term dependencies, laying the foundation for subsequent speech separation models. The dual-path approach enhances the model's resolution of speaker characteristics over longer time sequences, optimizing it for complex auditory scenes.

**DPTNet** (Chen et al., 2020a): Built on DPRNN, it incorporates a multi-head attention mechanism, further enhancing the model's ability to capture long-term contextual information and improving separation performance in complex speech scenarios. As one of the earliest works to introduce the concept of attention into the speech separation, it explores the application of attention in learning complex contextual correlations under the long-term characteristics of the speech domain.

**SuDORM-RF** (Tzinis et al., 2020): A time-domain speech separation model formed by stacking multiple UNet networks, it uses a refinement strategy to progressively optimize separation performance, improving the model's ability to handle complex signals. This model uses UNet to model different resolutions, and each additional UNet layer refines the separation detail, offering better clarity and fidelity in the final audio output.

**A-FRCNN** (Hu et al., 2021): Building on SuDORM-RF, it introduces cross-layer connections and parameter-sharing mechanisms, improving the model's capacity and parameter efficiency, leading to more precise speech separation. As an asynchronous update scheme, its top-down mechanism enables the model to better model multi-scale temporal information and achieve better results with fewer parameters.

**SKiM** (Li et al., 2022d): Incorporates the reuse of hidden states and cell states in DPRNN's inter-block modeling, strengthening contextual modeling ability and enabling the model to better handle long-term dependencies. In addition, SKiM also maintains the causal modeling ability of the model well, enabling it to handle real-time tasks with low latency and good performance.

**TDANet** (Li et al., 2023): Simplifies redundant connections in A-FRCNN and introduces a top-down attention mechanism to selectively extract multi-level acoustic features, further improving processing efficiency. The introduction of the attention module further enhances the model's ability to extract complex correlations of temporal information at different scales.

**BSRNN** (Luo & Yu, 2023b): Proposes a frequency-band splitting method, modeling both within frequency bands and across the time dimension using BLSTM, marking the first application of band-splitting methods in speech separation. This unique approach allows for more granular control over the frequency components, with better performance in tasks that rely more on fine frequency features, such as music separation tasks.

**TF-GridNet** (Wang et al., 2023): Extends DPRNN by modeling in both the time and frequency dimensions and integrates information from both dimensions using a multi-head attention module, leading to more refined and comprehensive speech separation. Compared to previous attention-based speech separation works, this work models in both time and frequency domains and reduces the length load of the attention module, making it more conducive to capturing contextual relationships.

**Mossformer** (Zhao & Ma, 2023): Combines convolution and self-attention in a gated, single-head Transformer architecture, utilizing full self-attention within local blocks and a linearized, low-cost self-attention mechanism for global modeling, significantly improving separation performance. The model also utilizes an attention-gathering mechanism with simplified single head self-attention. The model achieved good performance by integrating multiple modules.

**Mossformer2** (Zhao et al., 2024): Enhances Mossformer by introducing a feed-forward sequential memory network (FSMN) recurrent module, improving long-term dependency modeling and further boosting speech separation effectiveness. These FSMN blocks are mainly enhanced by using gated convolution units (GCUs) and dense connections. In addition, bottleneck layers and output layers have been added to control the flow of information. Based on the above content, the model integrates the loop module into the MossFormer framework, providing the ability to model remote, coarse-scale dependencies and fine-scale loop patterns.

## D.3   REAL-WORLD DATASET DETAILS

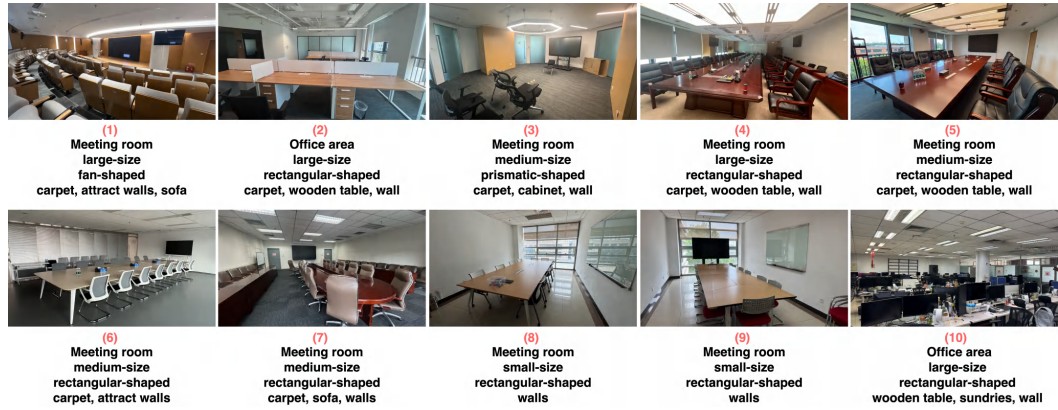

Figure 6: Recording audio using realistic scene images. These images show the layout of the actual physical environment used to record audio.

To evaluate the gap between the SonicSet dataset and real-world conditions, as well as the model's transferability to real-world scenarios, we collected a small-scale real-world moving sound source dataset, namely RealSEP and repeated the relevant experiments. First, we randomly selected 30 audio samples from 10 scenes in the SonicSet validation set, totaling 5 hours of audio. All data was clean, without reverberation or noise. During recording, one participant used the 2023 MacBook Pro's speakers to play audio while moving around randomly within the same scene to obtain the ground-truth audio. In addition, we used the same steps to play environmental and music noise from fixed positions to obtain noise data. The audio and noise were recorded using a Logitech Blue Yeti Nano omnidirectional microphone fixed in position, with a recording sample rate of 16 kHz and 32-bit depth. After recording, we clipped the audio and noise based on the recorded start and stop positions to ensure alignment with the original files. We then recorded another real-world dataset in 10 similar real-world scenes using the corresponding original audio. Finally, we mixed the data

using the same method as SonicSet to construct a dataset for testing model performance. The layout of the real-world scenes is illustrated in Figure 6.

In addition, we have constructed a dialogue speech dataset called *HumanSEP*, which contains recordings of 16 speakers in 6 different scenarios. Specifically, in each scenario, two speakers each randomly read aloud 15 sentences selected from the LibriSpeech dataset (Panayotov et al., 2015), with an average length of 4–6 seconds. To simulate real-world usage scenarios, the speech data was randomly interspersed with environmental noise. All audio was recorded using a fixed-position Logitech Blue Yeti Nano omnidirectional microphone with recording parameters set to a 16 kHz sampling rate and 32-bit depth. The dataset was also designed to include varying degrees of overlap between speakers to increase its diversity and challenge.

To evaluate the generalization performance of the models trained on SonicSet, we used the commonly used speech separation test datasets LibriMix (Cosentino et al., 2020) and LRS2 (Li et al., 2023). For the LibriMix dataset, we used the two-speaker mixed test set with a 16 kHz sampling rate. For the LRS2 dataset, we used the test set consistent with the (Li et al., 2023).

## D.4 TRAINING OBJECT

For all benchmark models in the speech separation, we employed the permutation invariant training (PIT) method (Hershey et al., 2016) to select the best output permutation $P \in \mathcal{P}_M$ (where $\mathcal{P}_M$ is the set of all $M$ permutations) to minimize the negative SNR adaptively. Specifically, SNR loss $\mathcal{L}_{\text{SNR}}$ can be defined as a negative SNR:

$$\mathcal{L}_{\text{SNR}}(\boldsymbol{x}_i, \bar{\boldsymbol{x}}_i) = -10 \log_{10} \left( \frac{\|\boldsymbol{x}_i\|^2}{\|\boldsymbol{x}_i - \bar{\boldsymbol{x}}_i\|^2} \right), \tag{1}$$

In this way, the final loss function of the PIT method under the SNR optimization objective can be expressed as:

$$\mathcal{L}_{\text{PIT-SNR}} = \min_{P \in \mathcal{P}_M} \sum_{i=1}^{M} \mathcal{L}_{\text{SNR}}(\boldsymbol{x}_i, \bar{\boldsymbol{x}}_i) \tag{2}$$

## D.5 IMPLEMENTATION DETAILS

During training, since all model hyperparameters were originally configured for the WSJ0-2Mix dataset (Hershey et al., 2016) and suited for audio with an 8 kHz sampling rate, we doubled the window length and window shift of the encoder and decoder to adapt to the 16 kHz sampling rate datasets. Aside from this adjustment, all other hyperparameters remained unchanged. For training, we randomly sampled 3s audio clips from the training set. The batch size was set to 1, and the Adam optimizer (Kingma, 2014) was used with an initial learning rate of $1 \times 10^{-3}$. The learning rate was halved whenever the validation loss did not decrease for five consecutive epochs. We applied gradient clipping to limit the maximum L2 norm of the gradients to 5. The training ran for a maximum of 500 epochs, with early stopping applied if the validation loss did not improve for 10 consecutive epochs.

In the speech separation experiments, we used SonicSet, public datasets, and real-world datasets. Each mixed audio contained two different speakers, with a sampling rate of 16 kHz. The baseline models pre-trained on the SonicSet training set were trained using a dynamic augmentation strategy. Specifically, during training, we randomly selected a set of data, and two speakers were randomly chosen from three available audio tracks for mixing. Depending on the task requirements, either environmental noise or musical noise was then mixed accordingly. Finally, a 3s segment was randomly extracted from the 60s audio for model training.

For each benchmark model, we selected the best model based on its performance on the validation set for testing. In the SonicSet evaluation, we used the full test set and employed a 6s inference window with a 3s sliding window to process long audio sequences. For the LRS2-2Mix dataset, we used 2s audio segments for both training and testing. For the Libri2Mix dataset, we trained and tested the models using 3s audio segments. All experiments were conducted on a server equipped with $8 \times$ NVIDIA 4090 GPUs.

## D.6 EVALUATION METRICS' DETAILS

Scale-invariant signal-to-noise Ratio (SI-SNR) (Le Roux et al., 2019) and Signal-to-distortion ratio (SDR) (Vincent et al., 2006) are standard metrics for evaluating the ratio of speech to noise and distortion, effectively measuring a model's ability to suppress noise while preserving the original signal. NB-PESQ and WB-PESQ (Rix et al., 2001) are used to assess the perceptual quality of wideband signals, respectively, based on human auditory models, predicting subjective speech quality, which is particularly suitable for evaluating signals processed through separation and enhancement. Additionally, STOI (Taal et al., 2011) measures speech intelligibility in noisy environments, helping to assess the loss of speech clarity in complex conditions.

Furthermore, Word Error Rate (WER), a standard in automatic speech recognition (ASR) systems, measures the error rate between the recognized words and the ground truth, reflecting the intelligibility of the speech signal. In this study, we use Whisper-medium-en (Radford et al., 2023) as the ASR model to recognize the content of separated audio.

By employing these multi-dimensional evaluation metrics, we can comprehensively assess the model's performance in real-world applications.

| Method | NISQA ↑ | WER (%) ↓ |
|---|---|---|
| Conv-TasNet | 1.37/1.45/1.51 | 90.61/88.11/86.97 |
| DPRNN | 1.24/1.28/1.31 | 82.48/76.00/72.29 |
| DPTNet | 1.45/1.57/1.63 | 81.89/74.73/70.57 |
| SuDoRM-RF | 1.32/1.38/1.45 | 85.21/79.31/76.69 |
| A-FRCNN | 1.33/1.40/1.44 | 77.34/72.48/69.64 |
| TDANet | 1.31/1.37/1.54 | 86.61/83.40/75.73 |
| SKIM | 1.21/1.29/1.31 | 94.43/87.26/85.67 |
| BSRNN | 1.54/1.56/1.64 | 81.92/77.58/73.08 |
| TF-GridNet | 1.77/1.83/1.96 | 70.60/64.65/62.81 |
| Mossformer | 1.60/1.64/1.76 | 68.66/61.23/57.81 |
| Mossformer2 | 1.62/1.68/1.78 | 68.21/62.01/56.58 |

Table 10: Comparative performance evaluation of models trained on different datasets using the HuamnSEP dataset. The results are reported separately for "*trained on LRS2-2Mix*", "*trained on Libri2Mix*" and "*trained on SonicSet*", distinguished by a slash. The relative length is indicated below the value by horizontal bars.

## D.7 DISCUSSION OF DIFFERENT METHODS

The performance of different speech separation models varied significantly under environmental noise and music noise conditions, highlighting how architectural choices impact their effectiveness. Among RNN-based models, DPRNN (Luo et al., 2020) and BSRNN (Luo & Yu, 2023b) performed relatively poorly when handling music noise. This suggests that these models face limitations in processing complex background music, likely due to the frequency variations in music noise posing significant challenges for RNN-based architectures. While recurrent architectures like DPRNN and BSRNN are effective in modeling long-term dependencies through bidirectional LSTM networks, they may not capture intricate contextual relationships as efficiently, especially in the presence of complex noise sources like music.

In contrast, CNN-based models such as SuDoRM-RF (Tzinis et al., 2020), A-FRCNN (Hu et al., 2021), and TDANet (Li et al., 2023) demonstrated more balanced performance across environmental and music noise scenarios. Their advantages in spectral refinement and signal reconstruction, particularly when leveraging multiple U-Net blocks, enabled them to effectively address complex noise interference. The incorporation of cross-layer connections and parameter-sharing mechanisms in TDANet enhances parameter efficiency and increases the model's capacity without proportionally

increasing computational load, leading to more precise speech separation. This design choice strikes a balance between model complexity, parameter efficiency, and performance, making TDANet more suitable for real-world applications where computational resources are constrained.

Among Transformer-based models, DPTNet (Chen et al., 2020a), TF-GridNet (Wang et al., 2023), Mossformer (Zhao & Ma, 2023), and Mossformer2 (Zhao et al., 2024) showed the best overall performance, especially in handling complex music noise while maintaining high separation accuracy. This superiority can be attributed to the multi-head attention mechanism, which effectively captures long-range dependencies and complex frequency variations. Attention mechanisms enhance the ability to model long-range dependencies and complex contextual correlations within speech signals, contrasting with the limitations of recurrent architectures. In music noise scenarios, this mechanism leverages contextual information to significantly reduce signal distortion, producing separated speech that is both clear and intelligible.

Furthermore, models like TF-GridNet (Wang et al., 2023) and Mossformer (Zhao & Ma, 2023) not only excelled in SI-SNR and SDR scores but also demonstrated high intelligibility scores, indicating their proficiency in producing separated speech that maintains both clarity and naturalness. TF-GridNet's innovative approach extends the dual-path recurrent neural network by modeling in both time and frequency dimensions and integrating information through a multi-head attention module. This dual-domain modeling strategy effectively captures temporal and spectral information, leading to significant improvements in separation performance.

The trade-off between model complexity, parameter efficiency, and performance is a critical aspect highlighted in the analysis. While top-performing models with advanced attention mechanisms offer superior separation quality, their computational complexity may pose challenges for real-time deployment. Latency and resource requirements become crucial factors when implementing these models in live systems. Therefore, models like A-FRCNN, which optimize computational efficiency without compromising performance, may be more practical for applications where resources are limited.

## E  BENCHMARK II: SPEECH ENHANCEMENT

### E.1  PROBLEM DEFINITION

Typically, the speech enhancement problem can be expressed as follows: given an observed noisy speech signal $s \in \mathbb{R}^{1 \times T}$, where $s = x + n$, with $x \in \mathbb{R}^{1 \times T}$ being the target speech signal and $n \in \mathbb{R}^{1 \times T}$ representing background noise, the goal of speech enhancement is to recover an estimate $\bar{x}$ of the target speech signal from $s$. Specifically, the process begins by applying the STFT to map the time-domain signal $s$ into time-frequency domain features $E \in \mathbb{C}^{F \times T'}$. Then, a speech enhancement network extracts the target speech from the noisy mixed signal, producing estimated time-frequency domain features $\bar{E} = f(E)$. Finally, the Inverse STFT is applied to reconstruct $\bar{E} \in \mathbb{C}^{F \times T'}$ back into the time-domain waveform $\bar{x} \in \mathbb{R}^{1 \times T}$, thus completing the recovery of the target speech signal.

### E.2  BENCHMARK MODELS

**DCCRN** (Hu et al., 2020) is a speech enhancement model designed for complex-domain processing. It operates in the frequency domain by handling both the real and imaginary components of speech signals, leveraging the characteristics of complex signals. Additionally, DCCRN employs a recurrent neural network to capture temporal information, making it particularly effective in processing speech in complex noisy environments. The model's innovation lies in its combination of complex-domain processing and temporal feature extraction, providing robust support for improving speech clarity and intelligibility.

**Fullband** (Hao et al., 2021) processes the entire frequency band of the speech signal, typically operating directly on either the full spectrum in the time or frequency domain. These approaches are particularly well-suited for speech enhancement tasks that address full-band noise. Leveraging complete frequency information effectively improves speech intelligibility and quality, making them a strong choice for applications in diverse noisy environments. Their comprehensive processing

capability ensures that both spectral and temporal features are preserved, leading to more natural and clear enhanced speech outputs.

**FullSubNet** (Hao et al., 2021) is a speech enhancement model that effectively combines full-band and sub-band information. It begins by extracting global features at the full-band level, subsequently refining these features at the sub-band level. This dual-layer processing approach enhances the overall performance of speech enhancement, allowing for more precise noise reduction and improved clarity. By integrating information from both frequency scales, FullSubNet successfully addresses various noise conditions, leading to more intelligible and natural-sounding speech outputs.

**Fast-FullSubNet** (Hao & Li, 2022) is an accelerated version of FullSubNet, specifically optimized for real-time speech enhancement. This model reduces computational complexity and improves inference speed, allowing faster speech processing while maintaining high enhancement quality. By streamlining the architecture and leveraging efficient algorithms, Fast-FullSubNet addresses the demands of real-time applications, making it suitable for scenarios where quick responses are critical.

**FullSubNet+** (Chen et al., 2022b) is an improved version of FullSubNet that further optimizes the model architecture and parameters for more efficient handling of sub-band information. It incorporates advanced techniques, such as channel attention mechanisms, to boost performance, particularly in complex noise environments. By refining the processing strategies and utilizing complex spectrograms, FullSubNet+ achieves superior noise reduction and clarity compared to its predecessor.

**TaylorSENet** (Li et al., 2022a) is a speech enhancement model that utilizes the Taylor series expansion to represent speech signal features. By expanding these features as a Taylor series, the model captures subtle variations in the signal, which enhances its robustness when processing complex audio signals. This innovative approach allows TaylorSENet to improve the overall quality and intelligibility of speech in challenging acoustic environments.

**GaGNet** (Li et al., 2022c) introduces a novel approach that combines gating mechanisms and guided learning to enhance speech separation tasks. The gating mechanism plays a crucial role in controlling the flow of information within the model, allowing for more effective differentiation between target speech and background noise. Simultaneously, the guided learning mechanism leverages external prior knowledge to assist the model in improving its separation capabilities. This dual approach not only enhances the model's performance in challenging acoustic environments but also facilitates better intelligibility and clarity of the separated speech, making GaGNet a significant advancement in the field of speech processing.

**G2Net** (Li et al., 2022b) enhances computational efficiency and performance by incorporating both gating mechanisms and group convolution. The use of group convolution effectively reduces the number of parameters in the model, leading to lower computational costs. Meanwhile, the gating mechanism plays a critical role in helping the model selectively process and prioritize key speech features. This combination not only optimizes the performance model but also makes it suitable for real-time speech enhancement tasks, ensuring that it delivers high-quality outputs even in demanding acoustic environments.

**Inter-SubNet** (Chen et al., 2023) is a speech enhancement model that focuses on interaction processing among sub-bands. By decomposing sub-band features, it facilitates information exchange between different sub-bands, effectively addressing inter-band dependencies. This innovative approach enhances the model's ability to capture the complex relationships between sub-bands, leading to improved speech enhancement outcomes. Through this interaction, Inter-SubNet significantly boosts the quality and intelligibility of the enhanced speech, making it particularly effective in challenging acoustic environments where traditional methods may struggle.

### E.3    TRAINING OBJECT

For all benchmark models used in the speech enhancement tasks, we applied various objective functions as outlined in the original papers to optimize model performance. Specifically, the SI-SNR loss function was employed in the DCCRN model (Hu et al., 2020). For the Fullband Hao et al. (2021), FullSubNet Hao et al. (2021), FullSubNet+ Chen et al. (2022b), FullSubNet-Fast Hao & Li (2022), and Inter-SubNet Chen et al. (2023) models, we utilized the complex ratio mask (cRM) error in the frequency domain as the core loss metric. By computing the mean square error (MSE) between the complex mask of the enhanced speech and the ideal mask, we can accurately quantify the model's

performance in the time-frequency domain. This method not only restores the magnitude of the speech signal but also effectively preserves its phase information. The loss function is expressed as:

$$\mathcal{L}_{\text{Fullband}} = \text{MSE}(\bar{\boldsymbol{M}}_{\text{cRM}}, \boldsymbol{M}_{\text{cIRM}}), \tag{3}$$

where $\bar{\boldsymbol{M}}_{\text{cRM}} \in \mathbb{C}^{F \times T'}$ is the estimated complex ratio mask, and $\boldsymbol{M}_{\text{cIRM}} \in \mathbb{C}^{F \times T'}$ is the ideal complex ratio mask.

For the TaylorSENet (Li et al., 2022a) and GaGNet (Li et al., 2022c) models, we used an Euclidean loss function based on complex magnitude. This loss function not only considers the differences in the complex space but also incorporates magnitude information, enabling the model to capture subtle changes in the input speech signal more precisely. The objective function is given as:

$$\mathcal{L}_{\text{TaylorSENet}} = \alpha \cdot \mathcal{L}_{\text{c}} + (1 - \alpha) \cdot \mathcal{L}_{\text{m}}, \tag{4}$$

where $\alpha$ is the weighting factor that controls the trade-off between the complex error $\mathcal{L}_c$ and the magnitude error $\mathcal{L}_m$. $\mathcal{L}_c$ is the loss of the complex part, which is defined as:

$$\mathcal{L}_{\text{c}}(\bar{\boldsymbol{x}}, \boldsymbol{x}) = |\bar{\boldsymbol{x}}_{\text{r}} - \boldsymbol{x}_{\text{r}}|^2 + |\bar{\boldsymbol{x}}_{\text{i}} - \boldsymbol{x}_{\text{i}}|^2, \tag{5}$$

where $\bar{x}_{\text{r}}$ and $\bar{x}_{\text{i}}$ represent the real and imaginary parts of the predicted STFT, respectively, and $\boldsymbol{x}_{\text{r}}$ and $\boldsymbol{x}_{\text{i}}$ are the real and imaginary parts of the ground-truth STFT. $\mathcal{L}_m$ is the loss of the amplitude part, which is defined as:

$$\mathcal{L}_{\text{m}}(\bar{\boldsymbol{x}}, \boldsymbol{x}) = (|\bar{\boldsymbol{x}}| - |\boldsymbol{x}|)^2. \tag{6}$$

In G2Net (Li et al., 2022b), we employed the stagewise complex magnitude Euclidean loss. This loss function applies different weights to the estimated signals at various stages, allowing the model to converge quickly in the early stages while fine-tuning the output signal quality in the later stages. The formula is as follows:

$$\mathcal{L}_{\text{G2Net}} = \sum_{i=1}^{N} \alpha_i \cdot \left( \mathcal{L}_{\text{c}}(\bar{\boldsymbol{x}}_i, \boldsymbol{x}) + \mathcal{L}_{\text{m}}(\bar{\boldsymbol{x}}_i, \boldsymbol{x}) \right), \tag{7}$$

where $\alpha_i$ is the weight for each stage, and $N$ is the number of stages in the model.

### E.4 Real-world dataset details

To evaluate the gap between the SonicSet dataset and real-world conditions, and the model's transferability to real-world scenarios, we collected a small real-world speech dataset for speech enhancement called *HumanENH*, which contains recordings from 16 speakers covering six different scenarios. Each speaker read 10 sentences from the LibriSpeech dataset (Panayotov et al., 2015) in the specified scenarios, with each sentence lasting 4-6s. During recording, we randomly added different ambient noises, such as traffic noise, cafe background noise and office ambient noise, to simulate real-world usage scenarios. All audio was recorded using a fixed-position Logitech Blue Yeti Nano omnidirectional microphone with recording parameters set to a 16 kHz sampling rate and 32-bit depth. The dataset is designed to be diverse and challenging, facilitating the evaluation of model performance on speech enhancement tasks.

### E.5 Discussion of different methods

The performance variations among the different speech enhancement models can be attributed to their underlying architectures and processing strategies. Subband-splitting methods like FullSubNet (Hao et al., 2021), Fast-FullSubNet (Hao & Li, 2022), FullSubNet+ (Chen et al., 2022b), and Inter-SubNet (Chen et al., 2023) excelled across multiple evaluation metrics due to their ability to handle frequency band dependencies effectively. By splitting the speech signal into subbands and facilitating inter-subband interaction, these models can capture detailed frequency-specific features and relationships. This approach enhances their capability to suppress noise and improve speech clarity, as it allows the models to address both local and global spectral variations more precisely. The interaction between subbands enables better modeling of complex acoustic patterns, leading to superior performance in terms of speech intelligibility and quality.

| Method | NISQA ↑ | WER (%) ↓ |
|---|---|---|
| DCCRN | 2.01/2.08/1.59 | 72.91/62.19/52.16 |
| Fullband | 2.12/2.19/1.78 | 46.98/36.48/29.86 |
| FullSubNet | 2.23/2.31/1.81 | 69.64/55.71/47.13 |
| Fast-FullSubNet | 2.24/2.29/1.81 | 54.87/42.49/33.28 |
| FullSubNet+ | 2.33/2.39/1.79 | 70.57/62.52/55.23 |
| TaylorSENet | 2.66/2.68/1.40 | 38.70/30.12/24.13 |
| GaGNet | 2.57/2.61/1.54 | 39.88/33.65/24.94 |
| G2Net | 2.42/2.42/1.49 | 45.92/40.28/30.94 |
| Inter-SubNet | 1.76/1.87/1.84 | 63.36/54.53/42.07 |

Table 11: Comparative performance evaluation of models trained on different datasets using the HumanENH dataset. The results are reported separately for "*trained on VoiceBank-DEMAND*", "*trained on DNS Challenge*" and "*trained on SonicSet*", distinguished by a slash.

In contrast, models like TaylorSENet (Li et al., 2022a) and GaGNet (Li et al., 2022c) demonstrated strong performance in objective metrics related to noise reduction, such as SI-SNR and SDR, indicating their effectiveness in suppressing background noise. However, they slightly lagged in subjective metrics like MOS, which assess perceived speech quality, including aspects like reverberation and signal fidelity. The possible reason for this disparity is that while these models are adept at denoising, they might not preserve the naturalness of the speech signal as effectively. Their architectures may not fully capture the nuances of reverberation effects or may introduce artifacts that affect the overall listening experience, suggesting limitations in maintaining signal integrity during the enhancement process.

