# OpenReview forum: "SonicSim: A customizable simulation platform for speech processing in moving sound source scenarios"
_ICLR.cc/2025/Conference — ICLR 2025 Poster_

### Official Review · Reviewer_xZx6 · 2024-11-01

**Soundness:** 3
**Presentation:** 2
**Contribution:** 3
**Rating:** 6
**Confidence:** 4

**Summary:**

This paper presents SonicSim, a toolkit for generating realistic synthetic data in dynamic audio environments, specifically for moving sound sources. Built on Habitat-sim, this platform allows users to control variables like scene configurations, microphone types, and source movements. Using SonicSim, the authors created SonicSet, a new dataset to test models for speech separation and enhancement in several conditions. Results suggest that models trained on SonicSet perform better in real-world settings than those trained on other synthetic datasets.

**Strengths:**

1. SonicSim provides a customizable environment where users can adjust elements like scene layouts, source paths, and microphone setups.
2. SonicSim and SonicSet offer a promising step toward bridging the gap between synthetic and real-world audio data, which addresses current limitations in data for training speech front-end models.
3. The paper presents well-organized experiments, testing a variety of speech separation and enhancement models with a range of evaluation metrics like SI-SNR, SDR, and WER. These thorough experiments help demonstrate the value of SonicSet in preparing models for real-world conditions.
4. I appreciate the authors’ effort to make their work reproducible by providing open-source code and dataset.

**Weaknesses:**

1. I find the real-world testing too limited, as it relies heavily on the RealMAN dataset for speech enhancement (Section 5.5). Expanding testing to include other real-world data, perhaps by offering qualitative results on some in-the-wild examples (where speech is from actual conversations rather than recorded playback), would give a more comprehensive picture of how models trained on SonicSet generalize. That said, RealMAN alone may not be enough to fully support the claims of SonicSet’s real-world effectiveness, and an additional perceptual study would be highly beneficial.
2. Although the authors claim that SonicSim improves realism in reverberation modeling, there isn’t enough quantitative or qualitative support for this. I would recommend directly comparing SonicSim’s synthetic RIRs to real RIRs to demonstrate this realism. This type of comparison could show how much the enhanced realism of SonicSim truly benefits model performance.
3. While the paper covers a range of models for speech separation and enhancement, it’s unclear why these specific models were chosen. I think the authors should consider discussing a bit why one model is better than the other, and including more analysis to clarify how well SonicSim supports different model types.
4. SonicSet uses fixed LUFS levels for environmental and musical noise, which may limit how robust models are to different real-world noise levels. Adding tests with varied noise levels (like signal-to-noise ratios (SNRs)) could help demonstrate SonicSet’s ability to handle diverse noise conditions. I would also be interested to see how varying the noise characteristics affects model performance.
5. It’s not clear to me how flexible SonicSim is regarding scene layouts and material properties. It would help if the authors explained what can be adjusted versus what is fixed. This would allow readers to understand any limits in SonicSim’s adaptability.
6. Moving key details (training details, dataset splits, and evaluation metrics) into the main text would improve readability and reduce the need for readers to frequently switch sections, especially for understanding the experimental setup. The current Introduction section feels overly detailed and could be streamlined (reducing adverb use would improve clarity).

**Questions:**

See weaknesses. I'd be happy to increase my rating if the authors address the weaknesses.

---

> ### Author Response · Authors · 2024-11-21
>
> **Q1: Expanding testing to include other real-world data, perhaps by offering qualitative results on some in-the-wild examples (where speech is from actual conversations rather than recorded playback), would give a more comprehensive picture of how models trained on SonicSet generalize**
>
> **A1:** We sincerely thank the reviewer for their suggestion. We chose ReaLMAN as the real-world speech enhancement dataset because it is the only dataset that features recordings of moving sound sources, making it particularly suitable for testing speech enhancement baseline models trained on the SonicSet dataset.
>
> Following your recommendation, we re-recorded speech data from 16 speakers across six different scenarios, tailored for both speech enhancement and separation tasks. For the speech enhancement task, each speaker read 10 sentences from the LibriSpeech dataset in a given scenario, with varying environmental noises added randomly during the recording process. For the speech separation task, pairs of speakers in each scenario read 15 sentences from the LibriSpeech dataset at random intervals, also with randomly added environmental noise. These recordings included varying overlap rates between speakers. These data will be made available for subsequent academic research.
>
> Using the methods described above, we constructed a new real-world recorded dataset （HumanSEP and HumanENH）and re-evaluated 19 baseline models (including both speech separation and enhancement models). In this experiment, we use NISQA and Whisper-medium-en to compute WER as metrics since there are no real labels. We used this dataset to compare with speech separation and enhancement models pre-trained on different datasets, as shown in Tables A1 and A2. The experimental results show that the models pre-trained on the SonicSet dataset can be better generalized to real conversation data. We have added this part to the revised manuscript.
>
> > Table A1: Comparative performance evaluation of separation models trained on different datasets using the HumanSEP dataset. The results are reported separately for "trained on LRS2-2Mix", "trained on Libri2Mix" and "trained on SonicSet", distinguished by a slash.
>
> | Method        | NISQA (↑)            | WER (%) (↓)           |
> |---------------|----------------------|-----------------------|
> | Conv-TasNet   | 1.37 / 1.45 / 1.51  | 90.61 / 88.11 / 86.97 |
> | DPRNN         | 1.24 / 1.28 / 1.31  | 82.48 / 76.00 / 72.29 |
> | DPTNet        | 1.45 / 1.57 / 1.63  | 81.89 / 74.73 / 70.57 |
> | SuDoRM-RF     | 1.32 / 1.38 / 1.45  | 85.21 / 79.31 / 76.69 |
> | A-FRCNN       | 1.33 / 1.40 / 1.44  | 77.34 / 72.48 / 69.64 |
> | TDANet        | 1.31 / 1.37 / 1.54  | 86.61 / 83.40 / 75.73 |
> | SKIM          | 1.21 / 1.29 / 1.31  | 94.43 / 87.26 / 85.67 |
> | BSRNN         | 1.54 / 1.56 / 1.64  | 81.92 / 77.58 / 73.08 |
> | TF-GridNet    | 1.77 / 1.83 / 1.96  | 70.60 / 64.65 / 62.81 |
> | Mossformer    | 1.60 / 1.64 / 1.76  | 68.66 / 61.23 / 57.81 |
> | Mossformer2   | 1.62 / 1.68 / 1.78  | 68.21 / 62.01 / 56.58 |
>
> > Table A2: Comparative performance evaluation of enhancement models trained on different datasets using the HumanENH dataset. The results are reported separately for "trained on VoiceBank-DEMAND", "trained on DNS Challenge" and "trained on SonicSet", distinguished by a slash.
>
> | Method           | NISQA (↑)            | WER (%) (↓)           |
> |-------------------|----------------------|-----------------------|
> | DCCRN            | 2.01 / 2.08 / 1.59  | 72.91 / 62.19 / 52.16 |
> | Fullband         | 2.12 / 2.19 / 1.78  | 46.98 / 36.48 / 29.86 |
> | FullSubNet       | 2.23 / 2.31 / 1.81  | 69.64 / 55.71 / 47.13 |
> | Fast-FullSubNet  | 2.24 / 2.29 / 1.81  | 54.87 / 42.49 / 33.28 |
> | FullSubNet+      | 2.33 / 2.39 / 1.79  | 70.57 / 62.52 / 55.23 |
> | TaylorSENet      | 2.66 / 2.68 / 1.40  | 38.70 / 30.12 / 24.13 |
> | GaGNet           | 2.57 / 2.61 / 1.54  | 39.88 / 33.65 / 24.94 |
> | G2Net            | 2.42 / 2.42 / 1.49  | 45.92 / 40.28 / 30.94 |
> | Inter-SubNet     | 1.76 / 1.87 / 1.84  | 63.36 / 54.53 / 42.07 |

---

> > ### Author Response · Authors · 2024-11-21
> >
> > **Q2: directly comparing SonicSim’s synthetic RIRs to real RIRs to demonstrate this realism.**
> >
> > **A2:** We appreciate the reviewer’s suggestion. Due to the lack of hardware equipment for 3D scene scanning and reconstruction, we were unable to obtain data from identical real and simulated environments. To address this, we utilized the RIRs dataset collected by the Habitat-sim team in the same real and simulated locations within the Replica Dataset [1]. This dataset includes RIRs recorded at seven distinct positions.
> > The provided files contain calculations of RT60 values for both real and simulated RIRs. The results, presented in the Table A3, demonstrate that the acoustic characteristics of audio simulated on the Habitat-sim platform, on which SonicSim is based, closely approximate those of audio recorded in real environments. This validates the reliability of the Habitat-sim-based simulations for acoustic modeling.
> >
> > > Table A3. The RT60 values of different positions.
> >
> > |                  | 1     | 2    | 3    | 4    | 5    | 6    | 7    |
> > |------------------|-------|------|------|------|------|------|------|
> > | Real World       | 1.48  | 0.50 | 1.49 | 1.54 | 1.30 | 1.42 | 0.68 |
> > | Habitat-sim (SonicSim) | 1.66  | 0.57 | 1.61 | 1.62 | 1.60 | 1.68 | 0.73 |
> >
> > **Q3: While the paper covers a range of models for speech separation and enhancement, it’s unclear why these specific models were chosen. I think the authors should consider discussing a bit why one model is better than the other, and including more analysis to clarify how well SonicSim supports different model types.**
> >
> > **A3:** Thank you for your suggestion. We selected the speech separation and enhancement models based on the fact that they were state-of-the-art (SOTA) methods at the time of their publication on various datasets and were open source. For instance, the DPRNN model, a foundational architecture for subsequent SOTA models, was included as a baseline. We have added a comparison and analysis of the different speech separation and enhancement models to the revised manuscript. Below is a summary of the main content:
> >
> > **Speech Separation Models**
> >
> > The results of the experiment are shown in Table 4. The performance of different models varied under environmental noise and music noise conditions. Among RNN-based models, DPRNN and BSRNN performed relatively poorly when handling music noise. This suggests that these models face limitations in processing complex background music, likely due to the frequency variations in music noise posing significant challenges for RNN-based architectures.
> >
> > In contrast, CNN-based models such as SuDoRM-RF, A-FRCNN and TDANet demonstrated more balanced performance across environmental and music noise scenarios. Their advantages in spectral refinement and signal reconstruction, particularly when leveraging multiple U-Net blocks, enabled them to effectively address complex noise interference. Furthermore, A-FRCNN's use of cross-layer connections and parameter sharing further enhanced its modeling capabilities for music noise.
> >
> > Among Transformer-based models, DPTNet, TF-GridNet, Mossformer and Mossformer2 showed the best overall performance, especially in handling complex music noise while maintaining high separation accuracy. This superiority can be attributed to the multi-head attention mechanism, which effectively captures long-term dependencies and complex frequency variations. In music noise scenarios, this mechanism leverages contextual information to significantly reduce signal distortion.

---

> > > ### Author Response · Authors · 2024-11-21
> > >
> > > **Speech Enhancement Models**
> > >
> > > The results of the experiment are shown in Table 6. The performance variations among the different speech enhancement models can be attributed to their underlying architectures and processing strategies. Subband-splitting methods like FullSubNet, Fast-FullSubNet, FullSubNet+, and Inter-SubNet excelled across multiple evaluation metrics due to their ability to handle frequency band dependencies effectively. By splitting the speech signal into subbands and facilitating inter-subband interaction, these models can capture detailed frequency-specific features and relationships. This approach enhances their capability to suppress noise and improve speech clarity, as it allows the models to address both local and global spectral variations more precisely. The interaction between subbands enables better modeling of complex acoustic patterns, leading to superior performance in terms of speech intelligibility and quality.
> > >
> > > In contrast, models like TaylorSENet and GaGNet demonstrated strong performance in objective metrics related to noise reduction, such as SI-SNR and SDR, indicating their effectiveness in suppressing background noise. However, they slightly lagged in subjective metrics, which assess perceived speech quality, including aspects like reverberation and signal fidelity. The possible reason for this disparity is that while these models are adept at denoising, they might not preserve the naturalness of the speech signal as effectively. Their architectures may not fully capture the nuances of reverberation effects or may introduce artifacts that affect the overall listening experience, suggesting limitations in maintaining signal integrity during the enhancement process.
> > >
> > > These findings highlight the strengths and limitations of different models under varying conditions and provide a deeper understanding of their applicability to complex real-world scenarios.

---

> ### Author Response · Authors · 2024-11-21
>
> **Q4: Adding tests with varied noise levels (like signal-to-noise ratios (SNRs)) could help demonstrate SonicSet’s ability to handle diverse noise conditions.**
>
> **A4:** To showcase SonicSet's capability to handle diverse noise conditions, we adjusted the signal-to-noise ratio (SNR) of the test set noise to include levels of -5 dB, 0 dB, 5 dB, and 10 dB. We then evaluated both speech separation and enhancement models on these test sets. The results are presented in Tables A4 and A5.
>
> > Table A4. The performance comparison of different speech separation models under various signal-to-noise ratio (SNR) levels with environmental noise, where the SNR range is [-5 / 0 / 5 / 10].
>
> | Method        | SI-SNR (↑)                | WB-PESQ (↑)          | STOI (↑)                | MOS Overall (↑)        | NISQA (↑)              | WER (%) (↓)              |
> |---------------|---------------------------|-----------------------|-------------------------|-------------------------|-------------------------|--------------------------|
> | TF-GridNet    | 11.64 / 15.38 / 15.59 / 16.05 | 2.81 / 3.08 / 3.01 / 3.15 | 90.70 / 93.32 / 93.99 / 94.12 | 1.96 / 2.49 / 2.45 / 2.51 | 1.80 / 1.91 / 1.93 / 1.92 | 21.23 / 12.04 / 11.86 / 11.80 |
> | Mossformer    | 10.54 / 14.72 / 14.98 / 15.51 | 2.31 / 2.67 / 2.70 / 2.77 | 88.75 / 91.13 / 91.73 / 92.20 | 1.75 / 2.39 / 2.43 / 2.42 | 1.53 / 1.86 / 1.87 / 1.91 | 38.19 / 21.10 / 20.01 / 18.24 |
> | Mossformer2   | 10.61 / 14.84 / 15.19 / 15.40 | 2.32 / 2.83 / 2.91 / 2.95 | 89.28 / 91.79 / 92.24 / 92.87 | 1.92 / 2.40 / 2.45 / 2.41 | 1.62 / 1.89 / 1.83 / 1.90 | 30.12 / 19.51 / 18.78 / 17.60 |
>
> > Table A5. The performance comparison of different speech enhancement models under various signal-to-noise ratio (SNR) levels with environmental noise, where the SNR range is [-5 / 0 / 5 / 10].
>
> | Method           | SI-SNR (↑)           | WB-PESQ (↑)         | STOI (%) (↑)          | MOS Overall (↑)      | NISQA (↑)            | WER (%) (↓)          |
> |-------------------|----------------------|----------------------|-----------------------|-----------------------|----------------------|----------------------|
> | FullSubNet       | 7.44/9.48/9.99/12.32 | 2.38/2.48/2.70/2.97 | 86.95/90.44/90.73/93.03 | 2.37/2.54/2.67/2.90 | 2.16/2.13/2.24/2.24 | 32.89/20.01/19.54/15.12 |
> | Fast-FullSubNet  | 6.56/8.14/9.29/11.69 | 2.32/2.41/2.71/2.98 | 87.21/90.04/93.20/93.06 | 2.56/2.58/2.63/2.63 | 2.14/2.09/2.15/2.25 | 24.89/21.13/19.87/16.54 |
> | Inter-SubNet     | 6.28/10.34/10.92/11.68 | 2.29/2.61/2.85/2.97 | 86.67/91.87/91.55/93.37 | 2.55/2.62/2.64/2.62 | 2.11/2.12/2.18/2.23 | 29.67/18.83/17.43/16.01 |
>
>
> Generally, as the SNR increased (i.e., the noise level decreases), the performance metrics improve for all models. The experimental results across different SNR levels confirm that models trained on SonicSet perform robustly under varying noise conditions. This demonstrates SonicSet's effectiveness in preparing models for real-world applications where noise levels can vary significantly. By providing diverse noise scenarios during training and testing, we ensure that the models can generalize well and maintain high performance in both low and high SNR environments.
>
> **Q5: It’s not clear to me how flexible SonicSim is regarding scene layouts and material properties. It would help if the authors explained what can be adjusted versus what is fixed. This would allow readers to understand any limits in SonicSim’s adaptability.**
>
> **A5:** We have incorporated this content into the revised manuscript. Regarding scene layout, SonicSim, leveraging the capabilities of the Habitat-sim platform, can import a variety of 3D scenes and add additional objects within these scenes. However, users cannot modify the 3D scenes themselves, as they are fixed during the import process. In terms of material properties, SonicSim supports a wide range of materials and objects, and users can modify the following physical and categorical attributes of materials:
> 1. Absorption: Represents the fraction of radiation or energy absorbed by the material, with the remainder potentially being scattered or transmitted.
> 2. Scattering: Indicates the material’s ability to scatter waves, which impacts the propagation paths of sound waves or light.
> 3. Transmission: Defines the material's ability to allow radiation to pass through.
> 4. Labels: Categorize or label materials to help identify their physical attributes.
> 5. Damping: Represents the damping effect of the material, influencing energy dissipation.
> 6. Density: Refers to the material's mass per unit volume, impacting its interaction with sound and energy.
> These features provide users with the flexibility to simulate a diverse range of acoustic environments and scenarios, contributing to SonicSim's versatility and realism.

---

> > ### Author Response · Authors · 2024-11-21
> >
> > **Q6: Moving key details (training details, dataset splits, and evaluation metrics) into the main text would improve readability and reduce the need for readers to frequently switch sections, especially for understanding the experimental setup. The current Introduction section feels overly detailed and could be streamlined (reducing adverb use would improve clarity).**
> >
> > **A6:** Thank you for your suggestions. We have adjusted these contents in the revised manuscript to make it easier for readers.
> >
> > **References**
> >
> > [1] Chen C, Schissler C, Garg S, et al. Soundspaces 2.0: A simulation platform for visual-acoustic learning[J]. Advances in Neural Information Processing Systems, 2022, 35: 8896-8911.

---

> > > ### Comment · Reviewer_xZx6 · 2024-11-25
> > >
> > > Thank you for addressing my concerns and conducting additional experiments—I appreciate your effort. I’m happy to increase my score to MA.
> > >
> > > Here are a few suggestions to improve clarity:
> > > 1. Please place each Figure and Table at the top of the page using "t" instead of "h" in LaTeX, as they currently seem buried in text.
> > > 2. On your project page, please include qualitative results from different methods. Skip the metric scores there, as people tend to ignore them at first glance.

---

### Official Review · Reviewer_jUje · 2024-11-01

**Soundness:** 3
**Presentation:** 3
**Contribution:** 3
**Rating:** 6
**Confidence:** 3

**Summary:**

The paper introduces SonicSim, a customizable simulation platform designed to generate synthetic data for moving sound source scenarios, addressing limitations in real-world and existing synthetic datasets for speech separation and enhancement models. Built on Habitat-sim, SonicSim enables multi-level adjustments across scene, microphone, and source levels, providing flexibility for diverse scenarios. Using SonicSim, the authors created SonicSet, a benchmark dataset derived from Librispeech, FSD50K, FMA, and Matterport3D to evaluate models under moving sound conditions. Additionally, real-world data comparisons validate SonicSim's synthetic data, showing promising generalization to real-world scenarios.

**Strengths:**

The paper offers a toolkit for generating customizable synthetic data for speech processing in moving sound source scenarios, an area where realistic data is scarce. SonicSim stands out for its versatility in multi-level adjustments and the breadth of scenarios it enables. The SonicSet dataset, derived from a combination of diverse sources, is a method for benchmarking models in this field. The authors also make an effort to bridge the synthetic-real data gap by comparing SonicSet's synthetic data with real-world datasets, which enhances the reliability of SonicSim as a training and evaluation tool. Evaluations against evaluation metrics are pretty solid.

**Weaknesses:**

One potential limitation is the lack of human evaluation MOS metrics for quality measurement. Existing metrics like PESQ and DNSMOS are used, but the paper lacks inclusion of more advanced full-reference, as well as no-reference models like NISQA or SQAPP. These metrics could provide a statistically significant, nuanced understanding of synthetic data quality versus similarity to real-world audio, strengthening the claims around SonicSim's effectiveness.

**Questions:**

1. Have the authors considered implementing advanced full-reference, as well as no-reference MOS metrics such as NISQA or SQAPP, or even non-matching reference metrics? These could enhance the quality assessment of synthetic data and offer more insight into the real-synthetic data gap.

2. How well does SonicSim perform in dynamically noisy environments, where background sounds change over time? Would it be feasible to test in such conditions to increase practical applicability?

---

> ### Author Response · Authors · 2024-11-21
>
> **Q1: Have the authors considered implementing advanced full-reference, as well as no-reference MOS metrics such as NISQA or SQAPP, or even non-matching reference metrics?**
>
> **A1:** Thanks to the reviewer for suggestion. We have included the NISQA evaluation metric in our speech separation and enhancement benchmarks to assess the quality of the models, as shown in the Tables 2-6 and 10-11 of the paper.
>
> **Q2: How well does SonicSim perform in dynamically noisy environments, where background sounds change over time? Would it be feasible to test in such conditions to increase practical applicability?**
>
> **A2:** It is important to clarify that each background noise sample in our dataset includes diverse content that changes over time. For instance, a single noise sample may contain sounds such as doors opening, footsteps, and walking, which occur at different moments within the same noise sequence. This design ensures that the simulated noise scenarios closely resemble real-world environments, thereby enabling better generalization to real-world test data.

---

> > ### Comment · Reviewer_jUje · 2024-12-02
> >
> > Thank you for addressing my concerns, I acknowledge the answers, but would still like to keep the same score. Thank you!

---

### Official Review · Reviewer_ADhE · 2024-11-03

**Soundness:** 3
**Presentation:** 3
**Contribution:** 3
**Rating:** 8
**Confidence:** 4

**Summary:**

SonicSim provides a new data simulator for moving sound sources for researchers working on speech separation and speech enhancement.  SonicSet is constructed using SonicSim using LibriSpeech, FreesoundDataset50k, and FreeMusicArchive, and 90 scenes from Matterport3D.

Using 9 speech separation models the authors show improvements when training on Voice-DEMAND, DNS Challenge, and SonicSet, and tested on the RealMAN test set, which shows significant improvement using SonicSet.

**Strengths:**

For researchers doing speech separation and enhancement with moving sound sources SonicSim and SonicSet will be valuable tools to create training data and training models. It fills an important gap in speech separation and enhancement dataset simulation.

**Weaknesses:**

SonicSim is constrained by the level of detail in 3D scene modeling.

For evaluation a subjective rating could have been used as done in DNS Challenge 2023 which would be more convincing than the objective metrics used (other than WER which is excellent).

**Questions:**

What is the 5-hour dataset described in C.5 called (please give it a name - it is not SonicSet). Also why was a MacBook Pro used to play back speech? That seems like a very poor choice to simulate a mouth. Why not use an artificial mouth instead, which mimics the response of a human speaker (frequency response and directionality)?

How can new microphone types be added?

Table 5: Says DNSMOS is used but it looks like SIGMOS used also? Please clarify which model MOS Sig, MOS BAK, and MOS Overall come from.

Table 6: Why are SI-SNR and WB-PESQ results much lower for Inter-SubNet than in the Inter-SubNet paper?

---

> ### Author Response · Authors · 2024-11-21
>
> **Q1: SonicSim is constrained by the level of detail in 3D scene modeling.**
>
> **A1:** We agree with your observation, and we have discussed this issue in the limitations section. The challenge arises because low-fidelity 3D scene modeling often introduces significant voids or gaps, which prevent rays in the audio simulation process from undergoing reflections, refractions, or diffraction. This, in turn, leads to inaccuracies in the simulated RIR. One potential solution is to smooth and refine the coarse 3D scene, thereby filling the voids in the mesh. This approach can mitigate the impact of this issue and improve the accuracy of the simulation.
>
> **Q2: For evaluation a subjective rating could have been used as done in DNS Challenge 2023 which would be more convincing than the objective metrics used**
>
> **A2:** In this study, we employed evaluation metrics commonly used in speech separation and enhancement research. These objective metrics provide a reliable reflection of model performance. While human evaluations can offer insights from a broader perspective, conducting them for our benchmarks would be prohibitively costly and time-intensive. Specifically, our evaluation involves 19 baseline models (spanning both speech separation and enhancement tasks) under two noise types. Assuming a test set comprising 40 samples of 60 seconds each and evaluations from 100 individuals, the process would require approximately 760 hours, which is beyond our feasible resources in terms of both time and cost.
>
> To address this limitation, we have supplemented our evaluations with DNSMOS and NISQA to assess differences among speech separation baseline models (see Tables 2-4 and 10 in the paper). For the enhancement task, we have included NISQA to evaluate performance differences among speech enhancement baselines (see Tables 5,6 and 11 in the paper). These additions ensure comprehensive and practical evaluations within our constraints.
>
> **Q3: What is the 5-hour dataset described in C.5 called (please give it a name - it is not SonicSet).**
>
> **A3:** We have named this dataset RealSEP in the revised manuscript to distinguish it from SonicSet.

---

> > ### Author Response · Authors · 2024-11-21
> >
> > **Q4: Also why was a MacBook Pro used to play back speech? That seems like a very poor choice to simulate a mouth. Why not use an artificial mouth instead, which mimics the response of a human speaker (frequency response and directionality)?**
> >
> > **A4:** The reason for using a MacBook Pro is that its speakers provide clearer output and superior audio quality, which helps ensure the quality of the recorded audio to a certain extent. During the rebuttal, we supplemented our experimental results by recording audio data from direct conversations between individuals. Specifically, we selected 16 speakers (12 men and 4 women, 23-28 years old) and used textual content from the LibriSpeech dataset for the conversations. The audio was collected across six scenarios, each containing 40 noisy and conversational audio samples of average lengths between 3 and 6 seconds. These datasets are named HumanSEP for the speech separation task and HumanENH for the speech enhancement task.
> >
> > We then evaluated the NISQA and WER results for speech separation and enhancement models trained on different datasets, as summarized in Tables A1 and A2. The experimental results demonstrate that the simulated audio data used in our study generalizes better to real-world environments, supporting the effectiveness of our approach. We have added this part to the revised manuscript.
> >
> > > Table A1: Comparative performance evaluation of separation models trained on different datasets using the HumanSEP dataset. The results are reported separately for "trained on LRS2-2Mix", "trained on Libri2Mix" and "trained on SonicSet", distinguished by a slash.
> >
> > | Method        | NISQA (↑)            | WER (%) (↓)           |
> > |---------------|----------------------|-----------------------|
> > | Conv-TasNet   | 1.37 / 1.45 / 1.51  | 90.61 / 88.11 / 86.97 |
> > | DPRNN         | 1.24 / 1.28 / 1.31  | 82.48 / 76.00 / 72.29 |
> > | DPTNet        | 1.45 / 1.57 / 1.63  | 81.89 / 74.73 / 70.57 |
> > | SuDoRM-RF     | 1.32 / 1.38 / 1.45  | 85.21 / 79.31 / 76.69 |
> > | A-FRCNN       | 1.33 / 1.40 / 1.44  | 77.34 / 72.48 / 69.64 |
> > | TDANet        | 1.31 / 1.37 / 1.54  | 86.61 / 83.40 / 75.73 |
> > | SKIM          | 1.21 / 1.29 / 1.31  | 94.43 / 87.26 / 85.67 |
> > | BSRNN         | 1.54 / 1.56 / 1.64  | 81.92 / 77.58 / 73.08 |
> > | TF-GridNet    | 1.77 / 1.83 / 1.96  | 70.60 / 64.65 / 62.81 |
> > | Mossformer    | 1.60 / 1.64 / 1.76  | 68.66 / 61.23 / 57.81 |
> > | Mossformer2   | 1.62 / 1.68 / 1.78  | 68.21 / 62.01 / 56.58 |
> >
> > > Table A2: Comparative performance evaluation of enhancement models trained on different datasets using the HumanENH dataset. The results are reported separately for "trained on VoiceBank-DEMAND", "trained on DNS Challenge" and "trained on SonicSet", distinguished by a slash.
> >
> > | Method           | NISQA (↑)            | WER (%) (↓)           |
> > |-------------------|----------------------|-----------------------|
> > | DCCRN            | 2.01 / 2.08 / 1.59  | 72.91 / 62.19 / 52.16 |
> > | Fullband         | 2.12 / 2.19 / 1.78  | 46.98 / 36.48 / 29.86 |
> > | FullSubNet       | 2.23 / 2.31 / 1.81  | 69.64 / 55.71 / 47.13 |
> > | Fast-FullSubNet  | 2.24 / 2.29 / 1.81  | 54.87 / 42.49 / 33.28 |
> > | FullSubNet+      | 2.33 / 2.39 / 1.79  | 70.57 / 62.52 / 55.23 |
> > | TaylorSENet      | 2.66 / 2.68 / 1.40  | 38.70 / 30.12 / 24.13 |
> > | GaGNet           | 2.57 / 2.61 / 1.54  | 39.88 / 33.65 / 24.94 |
> > | G2Net            | 2.42 / 2.42 / 1.49  | 45.92 / 40.28 / 30.94 |
> > | Inter-SubNet     | 1.76 / 1.87 / 1.84  | 63.36 / 54.53 / 42.07 |

---

> > > ### Author Response · Authors · 2024-11-21
> > >
> > > **Q5: How can new microphone types be added?**
> > >
> > > **A5:** Habitat-sim natively supports mono, stereo, and spatial audio. Building on this foundation, we have extended its capabilities to include linear and circular microphone arrays. The implementation is as follows:
> > > First, we define multiple single-channel receivers corresponding to the number of microphones in the array. Next, we configure the spatial relative coordinates of these single-channel receivers based on the desired array shape. Finally, we bind these receivers together and adjust the absolute coordinates to set the position of the microphone array within the environment.
> > > Through this process, users can easily modify and customize the microphone array type. We have encapsulated these steps into a high-level API, enabling users to seamlessly switch between pre-defined microphone types or define new custom microphone configurations with minimal effort.
> > >
> > > **Q6: Please clarify which model MOS Sig, MOS BAK, and MOS Overall come from.**
> > >
> > > **A6:**  We sincerely thank the reviewer for pointing out this issue. In Table 5, we used DNSMOS as the evaluation metric because the DNS Challenge dataset primarily employs DNSMOS for assessment. In Table 6, we used SigMOS for evaluation.
> > >
> > > **Q7: Why are SI-SNR and WB-PESQ results much lower for Inter-SubNet than in the Inter-SubNet paper?**
> > >
> > > **A7:** In Table 6, we use SonicSet as the test set, while the original paper uses the DNS Challenge dataset. Our focus is more on enhancing mobile sound sources, which poses additional challenges and may lead to performance degradation.

---

> > > ### Comment · Reviewer_ADhE · 2024-11-23
> > > **MacBook vs artificial mouth**
> > >
> > > I don’t think you addressed my comment about why a MacBook is a poor choice to simulate artificial mouth. What data do you have to show “its speakers provide clearer output and superior audio quality”?
> > >
> > > The biggest concern I have using a MacBook is the directivity response not close to a human mouth, which is highly directional. There are good reasons people use artificial mouths like the below and not studio quality speakers or MacBooks.
> > >
> > > https://www.bksv.com/en/transducers/simulators/ear-mouth-simulators/4227

---

> > > > ### Author Response · Authors · 2024-11-23
> > > > **Response to Comment on Using MacBook Pro vs. Artificial Mouth for Speech Simulation**
> > > >
> > > > We completely agree with your assessment that using a MacBook Pro to play back audio and record it with a microphone does not accurately represent human speech and cannot substitute for human voices. The artificial mouth hardware you referenced offers a much closer approximation of human speech characteristics than the MacBook Pro. Unfortunately, we currently do not have access to such equipment to experiment.
> > > >
> > > > To address this limitation, during the rebuttal period, we collected new real-world data (as detailed in **A4**), directly recording speech from different individuals reading English content. The experimental results demonstrate that the model trained on SonicSet achieves better generalization capabilities.
> > > > When we mentioned that the MacBook Pro provides clear output (the official website promotes high-fidelity speakers), it was solely to clarify our intent to minimize sound distortion during recording, not to contradict your valid point about the limitations of using such a setup.
> > > >
> > > > We appreciate your insightful feedback and the reference to the artificial mouth, which we acknowledge would have been a more suitable choice for simulating human speech.

---

### Official Review · Reviewer_h9Ga · 2024-11-04

**Soundness:** 2
**Presentation:** 3
**Contribution:** 2
**Rating:** 6
**Confidence:** 4

**Summary:**

The paper proposes a system and dataset for stimulating moving source sound data. The system is built using Habitat-sim which takes in a 3D scene and stimulates audio for receivers and transmitters placed at different locations. Simulated data for speech separation and speech enhancement problems are created, which are then used to benchmark performances of several state-of-the-art models for these problems. The results show that training models on simulated data can lead to better results on real-world dynamic audio.

**Strengths:**

– Simulating moving sound sources is indeed challenging and there are many good solutions for this problem. The paper relies on Habitat-sim and is a good attempt at improving simulation of moving sound sources.

– From audio perspective, the proposed system provides a reasonable degree of freedom to simulate in various conditions.

– The benchmarking of different speech separation and enhancement models is good – a variety of models for both tasks.

**Weaknesses:**

– The details of the moving source simulation could be improved a lot and is a major concern in the paper. Is the paper following  some known method ? Some mathematical description of what is going on in the moving source case is desirable, especially for description in Section 3.2.2 ? What exactly is SonicSim trajectory function “We then employed SonicSim’s trajectory function to calculate and generate a movement path for the sound source ” ?

– Is the speed of the moving source a factor in the whole simulation process ?

– Some ground truth validation of the simulations maybe helpful. For example, record data for a moving source in a real room and then use the 3D scene input of that room to simulate some data. How close are they ?

– While the benchmarking on speech enhancement and separation are nice, a more useful benchmarking might be RIR estimation ? There are plenty of recent works on neural room acoustics generation and I think this problem would provide more insightful picture of the simulation system.

– the paper mentions several challenges in audio simulations like obstacles, room geometry, room surface but none of them end up being a factor studied by the paper.

– some aspects of the audio generation settings like the microphone type, monoaural vs binaural might be good to showcase.

**Questions:**

Please see the questions above.

---

> ### Author Response · Authors · 2024-11-21
>
> **Q1: What exactly is SonicSim trajectory function "We then employed SonicSim’s trajectory function to calculate and generate a movement path for the sound source" ?**
>
> **A1:** The SonicSim trajectory function is built upon the Habitat-sim platform and computes potential trajectories between a specified starting point and endpoint. These trajectories are generated using Habitat-sim's path-finding API, specifically the `habitat_sim.ShortestPath()` method. Subsequently, SonicSim utilizes the trajectory to synthesize moving sound sources based on the method described in A2.
>
> **Q2: The details of the moving source simulation could be improved a lot and is a major concern in the paper. Is the paper following some known method ? Some mathematical description of what is going on in the moving source case is desirable, especially for description in Section 3.2.2 ?**
>
> **A2:** Thank you for raising this question. We employed the widely used method of smooth interpolation [1, 2] to transition between RIRs at different positions, ensuring that the audio effect does not exhibit abrupt changes due to rapid shifts in source position. To enhance the clarity of the moving sound source simulation process, we have added the following details in the revised manuscript.
> SonicSim selects the starting and endpoint positions of the sound source in a 3D space and then uses Habitat-sim's path-finding API to generate a navigable trajectory. The trajectory points are used to construct a smooth interpolation path and calculate the corresponding interpolation weights, enabling realistic spatial movement effects during audio synthesis.
> Assuming the source audio signal is $\mathbf{s}(t)$ with duration $T$, and the room impulse responses (RIRs) $\mathbf{h}_j^c(t)$ describe the transmission characteristics from each position $j$ to the receiver, where $j = 1, 2, \dots, N$ represents $N$ discrete positions and $c$ is the audio channel index, the process is as follows:
>
> 1. Convolution at Discrete Positions:
> The source signal is convolved with the RIR $\mathbf{h}_j^c(t)$ at each position $j$, resulting in the convolution response:
> $$
>    \mathbf{y}_j^c(t) = \mathbf{s}(t) * \mathbf{h}_j^c(t),
> $$
> where $*$ denotes the convolution operation.
>
> 2. Linear Interpolation Between Positions:
> Since moving sound sources transition smoothly between discrete positions, interpolation between the convolution results of adjacent positions is necessary. We introduce a linear interpolation weight $\alpha(t)$, which indicates the degree of transition between positions $j$ and $j+1$. The weight $\alpha(t)$ ranges from $0$ (fully at position $j$) to $1$ (fully at position $j+1$).
> The interpolation weights are calculated based on the Euclidean distance between the moving source's current position and the neighboring positions:
> $$
> \alpha(t) = \frac{d _ j - \text{dist}(\mathbf{r} _ j, \mathbf{r} _ t)}{d_j - d _ \{j+1\}}
> $$
> where $d_j$ and $d_{j+1}$ are the spatial distances between positions $j$ and $j+1$, and $\text{dist}(\mathbf{r}_j, \mathbf{r}_t)$ is the distance between the receiver's current position $\mathbf{r}_t$ and position $j$.
>
> 3. Weighted Average of Convolution Results:
> At each time step $t$, the interpolated audio signal is computed as the weighted average of the convolution results at adjacent positions:
> $$
> \mathbf{y}(t) = (1 - \alpha(t)) \cdot \mathbf{y} _ {\mathbf{i}(t)}^c(t) + \alpha(t) \cdot \mathbf{y} _ {\mathbf{i}(t)+1}^c(t),
> $$
> where $\mathbf{i}(t)$ denotes the position index at time $t$, $\mathbf{y} _ {\mathbf{i}(t)}^c(t)$ and $\mathbf{y} _ {\mathbf{i}(t)+1}^c(t)$ are the convolution results at positions $j$ and $j+1$, respectively, and $\alpha(t)$ is the interpolation weight.
>
> This approach ensures the synthesis of audio signals that reflect the smooth spatial movement of the sound source, providing a realistic simulation of moving sound sources.
>
> **Q3: Is the speed of the moving source a factor in the whole simulation process ?**
>
> **A3:** Thank you very much for your insightful question. We have included more details about speed in the revised manuscript. In SonicSet, we need to keep the audio length at a fixed length of 60s to facilitate our testing. In the simulation process, SonicSim adjusts the speed of movement based on the trajectory length to ensure compatibility with the fixed audio duration. For instance, the movement speed for a 10m-long trajectory is faster than that for a 5m-long trajectory. Since we randomly sample different 3D scenes during the simulation, the movement speed varies randomly, reflecting the diversity of real-world scenarios.

---

> > ### Author Response · Authors · 2024-11-21
> >
> > **Q4: Some ground truth validation of the simulations maybe helpful. For example, record data for a moving source in a real room and then use the 3D scene input of that room to simulate some data. How close are they ?**
> >
> > **A4:** Thank you very much for your suggestion. We do not have large-scale 3D modeling instruments to scan and recreate simulation scenes identical to real environments. But the Habitat-sim team has collected RIR reverberation data in both real and simulated scenes (Replica Dataset) [3]. This dataset is publicly available at [http://dl.fbaipublicfiles.com/SoundSpaces/real_measurements.zip], and it includes recordings of RIRs at seven different positions. The provided files include the RT60 values of real and simulated RIRs, as shown in Table A1. The results demonstrate that the acoustic characteristics of audio simulated in Habitat-sim are close to those of audio recorded in real environments.
> >
> > > Table A1. The RT60 values of different positions.
> >
> > |                  | 1     | 2    | 3    | 4    | 5    | 6    | 7    |
> > |------------------|-------|------|------|------|------|------|------|
> > | Real World       | 1.48  | 0.50 | 1.49 | 1.54 | 1.30 | 1.42 | 0.68 |
> > | Habitat-sim (SonicSim) | 1.66  | 0.57 | 1.61 | 1.62 | 1.60 | 1.68 | 0.73 |
> >
> > Additionally, we indirectly validated the authenticity of our simulations using a dataset containing 5 hours of real-world recordings. The experimental results in Tables 2, 3, and 5 of the paper indicate that models trained on datasets simulated with SonicSim demonstrate better separation performance on real-world data. These findings provide evidence for the realism and utility of our simulation process, bridging the gap between simulated and real-world scenarios.
> >
> > **Q5: While the benchmarking on speech enhancement and separation are nice, a more useful benchmarking might be RIR estimation ?**
> >
> > **A5:** Thank you for your suggestion. We investigated RIR estimation tasks during the rebuttal, which used room information, image information, and microphones and sound source locations to predict RIRs. Some methods include AV-RIR [4], AV-NeRF [5], Image2Reverb [6], and NAF [7]. RIR estimation is an important research direction, but our current focus is on the speech front-end under moving sound sources. This is because of the lack of sufficient data, and research progress in speech front-end tasks with moving sound sources has been slow. We hope SonicSim can provide more data to facilitate speech front-end tasks (such as separation and enhancement) under moving sound sources.

---

> ### Author Response · Authors · 2024-11-21
>
> **Q6: The paper mentions several challenges in audio simulations like obstacles, room geometry, room surface but none of them end up being a factor studied by the paper.**
>
> **A6:** We randomly selected 300 audio files from the Librispeech test-clean set. 200 of these were used to generate noisy mixtures containing two speakers, and the other 100 were used to generate noisy audio from a single speaker. The same audio files were used in both cases. To fix the scene, we chose a cuboid room with no obstructions but included room materials and fixed the positions of the microphone, sound source, and noise. In the experiment on occlusion, we used Habitat-sim's API `add_object` to place objects between the microphone and the sound source to simulate the occlusion in sound wave propagation. When studying the effect of room shape, we chose a cylindrical room from the Matterport3D dataset and fixed the relative positions of the microphone, sound source, and noise. In addition, we completely removed the root material information from the experiment to explore the effect of the material.
>
> We selected the two methods with the best speech separation and enhancement performance and verified the impact of different environmental factors on the model performance. The experimental results are shown in Tables A2 and A3. The results show that material has the most significant effect on model performance, while the impact of room shape is relatively small. However, all these factors have a certain degree of interference in model performance. Therefore, a simulated dataset using these environmental factors is generated to train the model and improve its generalization ability. This analysis has been added to Appendix A.
>
> > Table A2: Comparison of the performance of speech separation models for different environmental factors.
>
> | Separation Methods | With/Without Obstacles SDR (↑) | With/Without Obstacles WER (%) (↓) | Cylinder/Cuboid SDR (↑) | Cylinder/Cuboid WER (%) (↓) | With/Without Materials SDR (↑) | With/Without Materials WER (%) (↓) |
> |---------------------|-------------------------------|-------------------------------------|--------------------------|-----------------------------|----------------------------------|------------------------------------|
> | Mossformer          | 10.70 / 11.84               | 29.90 / 24.31                     | 11.18 / 11.84           | 27.17 / 24.31              | 11.84 / 10.12                  | 24.31 / 31.10                     |
> | Mossformer2         | 10.41 / 11.60               | 31.89 / 25.86                     | 10.78 / 11.60           | 30.54 / 25.86              | 11.60 / 9.87                   | 25.86 / 33.58                     |
>
> > Table A3: Comparison of the performance of speech enhancement models for different environmental factors.
>
> | Enhancement Methods | With/Without Obstacles SDR (↑) | With/Without Obstacles WER (%) (↓) | Cylinder/Cuboid SDR (↑) | Cylinder/Cuboid WER (%) (↓) | With/Without Materials SDR (↑) | With/Without Materials WER (%) (↓) |
> |----------------------|-------------------------------|-------------------------------------|--------------------------|-----------------------------|----------------------------------|------------------------------------|
> | FullSubNet           | 9.67 / 11.05                | 25.76 / 19.80                     | 10.56 / 11.05           | 21.94 / 19.80              | 11.05 / 10.23                  | 19.80 / 21.17                     |
> | Inter-SubNet         | 10.78 / 12.69               | 22.96 / 17.68                     | 11.50 / 12.69           | 19.30 / 17.68              | 12.69 / 11.26                  | 17.68 / 19.88                     |
>
> **Q7: Some aspects of the audio generation settings like the microphone type, monoaural vs binaural might be good to showcase.**
>
> **A7:** Thank you for your suggestion. We have added the hyperparameter settings for generating audio to the Appendix C. The hyperparameter settings include aspects such as microphone types, sound source starting and ending points, and microphone positions. Additionally, we have updated the code to provide a higher-level API that allows users to directly manipulate these hyperparameters, ensuring greater ease of use.

---

> > ### Author Response · Authors · 2024-11-21
> >
> > **References**
> >
> > [1] Slaney M, Covell M, Lassiter B. Automatic audio morphing. 1996 IEEE International Conference on Acoustics, Speech, and Signal Processing Conference Proceedings. IEEE, 1996, 2: 1001-1004.
> >
> > [2] Freeland F P, Biscainho L W P, Diniz P S R. Efficient HRTF interpolation in 3D moving sound. Audio Engineering Society Conference: 22nd International Conference: Virtual, Synthetic, and Entertainment Audio. Audio Engineering Society, 2002.
> >
> > [3] Chen C, Schissler C, Garg S, et al. Soundspaces 2.0: A simulation platform for visual-acoustic learning. Advances in Neural Information Processing Systems, 2022, 35: 8896-8911.
> >
> > [4] Ratnarajah A, Ghosh S, Kumar S, et al. Av-rir: Audio-visual room impulse response estimation. Proceedings of the IEEE/CVF Conference on Computer Vision and Pattern Recognition. 2024: 27164-27175.
> >
> > [5] Liang S, Huang C, Tian Y, et al. Av-nerf: Learning neural fields for real-world audio-visual scene synthesis. Advances in Neural Information Processing Systems, 2023, 36: 37472-37490.
> >
> > [6] Singh N, Mentch J, Ng J, et al. Image2reverb: Cross-modal reverb impulse response synthesis. Proceedings of the IEEE/CVF International Conference on Computer Vision. 2021: 286-295.
> >
> > [7] Luo A, Du Y, Tarr M, et al. Learning neural acoustic fields. Advances in Neural Information Processing Systems, 2022, 35: 3165-3177.

---

> > ### Comment · Reviewer_h9Ga · 2024-11-25
> > **Post rebuttal comment**
> >
> > Thank you for providing detailed answers. Additional details answers several questions I had. Increasing the score to reflect that.

---

### Author Response · Authors · 2024-11-21
**Hope the reviewers will take note of our response**

Dear Reviewers,

After submitting the initial comments, we incorporated your feedback into a revised version of our paper, performed some additional experiments as you requested, and wrote a response to address your main concerns.

We hope to interact with you during the discussion and potentially further improve the quality of our paper.

Thank you very much in advance.

Kind regards,

Authors

---

### Meta-Review · Area_Chair_WobG · 2024-12-21

**Metareview:**

This paper introduces SonicSim, a simulation platform built on Habitat-sim that generates synthetic data for speech separation and enhancement in scenarios with moving sound sources. While reviewers recognized the value of such an advanced simulation tool for these dynamic conditions, they initially raised concerns regarding missing details, clarifications, and results. The authors effectively addressed these concerns by providing further information on RIR interpolation, scene complexity, and by using newly recorded real speech data to validate model performance. These clarifications, along with new experimental results, including tests with varied noise levels and additional no-reference metrics, satisfied the reviewers' primary concerns.

Given the platform's value and the reviewers' final positive reviews (8, 6, 6, 6), I recommend acceptance. The authors should revise the paper to incorporate these new results and clarifications.

**Additional Comments On Reviewer Discussion:**

Initially, the reviewers requested further clarifications, additional results, and extended evaluations. The authors responded with further explanations and new experiments. After discussion, two reviewers (h9Ga and xZx6) raised their scores to 6, and overall ratings became unanimously positive.

---

### Decision · Program_Chairs · 2025-01-22

Accept (Poster)